# Post-Processing Approach for Distributive Fairness in Multi-Class Federated Learning

## Abstract

Distributive fairness is a critical concern in the application of Federated Learning (FL) to decision making. Three concepts of distributive fairness are recently considered important in FL: global, local group and client fairness. Global fairness addresses disparities among legally protected groups across the entire population. Local group fairness addresses disparities between protected groups within individual clients. Client fairness focuses on disparities across clients. These concepts of distributive fairness coexist in FL and achieving one does not guarantee the others. Most FL studies focus on only a single concept. In real-world applications, however, different stakeholders often require fairness from different perspectives simultaneously. Enforcing those fairness concepts inherently incurs an accuracy cost. This paper investigates that, for a given FL setup, the maximum achievable accuracy under various combinations of distributive fairness, i.e., all three, any two, or just one, depending on the application. We propose a post-processing algorithm that returns a model with the near-optimal accuracy while satisfying pre-specified fairness constraints. Experimental results show that our algorithm outperforms the current state of the art (SOTA) in terms of the fairness–accuracy tradeoff, computational and communication efficiency. Code is available on Github.

## 1 Introduction

Federated learning (FL) (McMahan et al., 2017) is a distributed machine learning framework that uses data collected from a group of *clients* to learn a global model that can be used by all clients. With the growing application of FL in finance (Long et al., 2020; Byrd & Polychroniadou, 2020; Nevrataki et al., 2023), hiring (Nguyen-Khanh et al., 2022; Zhang et al., 2023)and healthcare (Xu et al., 2021; Antunes et al., 2022; Chaddad et al., 2023), FL models are legally required to be fair, ensuring that they do not discriminate against different subpopulations. These subpopulations can be defined w.r.t. legally protected (a.k.a sensitive) attributes or geographic locations, which give rise to various concepts of distributive fairness in FL. This paper discusses applications in which different stakeholders may seek to achieve different distributive justice. We study the corresponding distributive fairness concepts and propose a framework that enforce them to a pre-specified level.

There are three distributive fairness (D-Fair) concepts considered critical in the FL literature: global group fairness (Abay et al., 2020; Du et al., 2021; Ezzeldin et al., 2023), local group fairness (Cui et al., 2021; Zhou & Goel, 2025) and client fairness (Li et al., 2019; Mohri et al., 2019). Global group fairness aligns with centralized group fairness (Hardt et al., 2016; Zafar et al., 2017b), which addresses disparities between sensitive and non-sensitive groups across the entire population. Local group fairness focuses on disparities between sensitive and non-sensitive groups within each individual client. Client fairness ensures that individual clients are not disadvantaged. These fairness concepts are distinct and achieving one fairness concept may not imply the others (Hamman & Dutta, 2023; Rychener et al., 2025). Fig. 1 (best viewed in color) explains how they differ. Global group fairness requires equality between protected groups over all population. Those groups are circled in red. Local group fairness ensures equality between protected groups within each client, which are circled in green. Client fairness ensures equality across clients, which are circled in blue.

Consider an FL system defined by the tuple $D = (X, A, C, Y)$ with distribution $F_D : \mathcal{X} \times \mathcal{A} \times \mathcal{C} \times \mathcal{Y} \to [0, 1]$, where, $x \in \mathcal{X}$ is the individual's *private profile*, $a \in \mathcal{A} = \{0, 1\}$ denotes the individual's *sensitive attribute*, $c \in \mathcal{C} = \{1, 2, \ldots, K\}$ specifies the *client* to

which the individual belongs and $y \in \mathcal{Y} = \{1, \cdots, N\}$ is the individual's *qualified outcome*. $\widetilde{Y} : \mathcal{X} \times \mathcal{A} \times \mathcal{C} \to \mathcal{Y}$ is an outcome predictor, a.k.a model. Global group fairness (under *Statistical Parity* (Dwork, 2006)) requires that the predictor's outcome $\widetilde{Y}$ is independent of the sensitive attributes $A$: $\widetilde{Y} \perp A$. Local group fairness requires independence between the outcome $\widetilde{Y}$ and the sensitive attributes $A$, conditioned on the client $C$: $\widetilde{Y} \perp A|C$. Finally, client fairness requires that the predictor's outcome $\widetilde{Y}$ is independent of the client identifier: $\widetilde{Y} \perp C$.

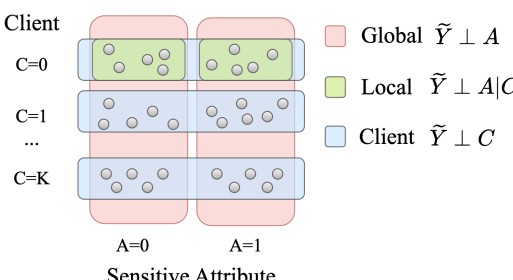

Figure 1: Global group fairness (red), local group fairness (green) and client fairness (blue).

Most FL studies (Abay et al., 2020; Zeng et al., 2021; Ezzeldin et al., 2023; Salazar et al., 2023; Hamman & Dutta, 2023; Makhija et al., 2024) focus on a single D-Fair concept. In real-world applications, however, different D-Fair concepts are required by different stakeholders. Consider several local hospitals serving as clients in a city (Linardos et al., 2022), these clients collaboratively train a model to determine whether an individual has a certain illness and qualifies for healthcare benefits. Hospitals are legally required (Congress, 2022) to ensure that medical decisions are not influenced by protected attribute (local group fairness). The model are also regulated to be fair between sensitive and non-sensitive groups citywide (global group fairness). Meanwhile, city leaders also want to ensure all geographically distinct neighborhoods have equal access to adequate healthcare, as neighborhoods that perceive themselves as neglected are less likely to engage in municipal governance (Salisu et al., 2023)(client fairness). More real-world applications that consider multiple fairness concepts are discussed in Appendix A.

Furthermore, while existing studies of fair FL (Li et al., 2019; Mohri et al., 2019; Abay et al., 2020; Zeng et al., 2021; Ezzeldin et al., 2023; Salazar et al., 2023; Makhija et al., 2024; Zhang et al., 2025) have primarily focused on binary-class settings, most real-world problems (Arunkumar & Karthigaikumar, 2017) are multi-class. Motivated by the fact that real applications are often multi-class problems and require different D-Fair concepts, this paper proposes a unified framework that addresses these concepts in multi-class FL. We formulate the problem of finding the optimal fair outcome predictor as a convex program, which presents the maximum achievable accuracy under given fairness constraints. Our framework is leads to a post-processing approach that first trains a model using `FedAvg` (McMahan et al., 2017) then enforces fairness through a linear program (LP), whose solution approximates the optimal fair predictor. We discuss related work on fair FL frameworks and post-processing techniques, followed by a summary of our contributions.

**Related Work:** (A) Fair Federated Learning: Existing methods that enforce fairness in FL fall into three categories: pre-processing, in-processing and post-processing. Pre-processing methods (Abay et al., 2020) enforce fairness by modifying the training dataset. It cannot support multiple fairness concepts as the global server cannot access to clients' data. In-processing methods (Mohri et al., 2019; Li et al., 2019; Rodríguez-Gálvez et al., 2021; Zeng et al., 2021; Du et al., 2021; Papadaki et al., 2022; Yue et al., 2023; Ezzeldin et al., 2023; Makhija et al., 2024; Rychener et al., 2025) modify the FL optimization algorithm. Li et al. (2019); Yue et al. (2023); Ezzeldin et al. (2023), for example, use dynamic aggregation weights. Mohri et al. (2019); Du et al. (2021); Papadaki et al. (2022) uses min-max training. These approaches complicate the FL training algorithm and increase communication and computational costs. Most of them lack convergence guarantees and do not provide optimal accuracy analysis under fairness constraints. Post-processing techniques (Zhang et al., 2025) transform a pre-trained model into a fair one. Existing post-processing framework only supports binary class. We summarize our **contributions regarding fairness in FL** as follows:

(1) We propose a framework that, for a given FL setup, determines the optimal accuracy under local, global and client fairness constraints. The framework leads to a simple post-processing algorithm. It is well-suited for real applications where multiple fairness concepts are simultaneously required.

(2) The post-processing algorithm preserves FedAvg convergence and thus has lower communication and computation costs than other fair FL framework.

(3) Our framework supports multi-class and all common fairness metrics.

(4) Experiments show it enforces fairness with significantly reduced communication and computation cost compared to SOTA.

(B) Post-processing techniques: Post-processing transform a pre-trained model to a fair model. Existing post-processing frameworks (Hardt et al., 2016; Chzhen et al., 2019; Denis et al., 2021; Gaucher et al., 2023; Xian et al., 2023; Xian & Zhao, 2024; Zhang et al., 2025; Zhou & Goel, 2025) primarily focus on centralized machine learning and cannot be directly applied in FL due to privacy concerns. Besides that, most post-processing frameworks focus on specific fairness metrics and problem settings. Jiang et al. (2020); Denis et al. (2021); Gaucher et al. (2023); Xian et al. (2023) only support Statistical Parity. Hardt et al. (2016); Chzhen et al. (2019); Zhang et al. (2025); Zhou & Goel (2025) only support binary-class settings. Among post-processing techniques, the most relevant to our work is (Hardt et al., 2016), which enforces fairness via a convex program optimizing model performance under fairness constraints. The feasible region of the convex program is defined by the Receiver Operating Characteristic (ROC) curve. The ROC curve is derived by varying the threshold on a pre-trained score function. Hardt et al. (2016) is limited to binary-class problem. We extend Hardt et al. (2016) to the multi-class setting. This extension is non-trivial. We make the following **technical contributions to ROC-based post-processing**:

(1) We formally defines the ROC surface w.r.t. the weighted score function (Def. 2.3). In Hardt et al. (2016), the ROC curve is derived by varying a threshold over a scalar score. Their framework does not apply to the multi-class case, where the pre-trained score is a high-dimensional vector.

(2) We defines the region under the ROC surface using supporting hyperplanes that separate this region from others (Def. 2.4, 2.5).

(3) We prove that the region under the ROC surface is convex. (Prop. 2.6).

(4) We prove that for any predictor $\widetilde{Y} : \mathcal{X} \times \mathcal{A} \times \mathcal{C} \to \mathcal{Y}$, the true positives lie within the region under the ROC surface (Prop. 2.6). Thus the feasible region of our convex program (Prop.3.1) includes all achievable true positives for any outcome predictor. Therefore, the solution to this convex program yields the optimal accuracy under fairness constraints.

## 2 PRELIMINARIES

### 2.1 NOTATIONS AND DEFINITIONS

Consider FL system is defined as $D = (X, A, C, Y)$, we are interested in selecting an *outcome predictor*, $\widetilde{Y} : \mathcal{X} \times \mathcal{A} \times \mathcal{C} \to \mathcal{Y}$ for a FL system $D$ that satisfies multiple distributive fairness concepts. We now formally define these concepts.

**Definition 2.1** *Let* $\epsilon_g, \epsilon_l, \epsilon_c \in [0, 1]^3$ *be the pre-specified fairness levels. The local group, global group, and client fairness are defined as follows:*

*(1) $\epsilon_g$-**Global Group Fairness:** The outcome predictor* $\widetilde{Y} : \mathcal{X} \times \mathcal{A} \times \mathcal{C} \to \mathcal{Y}$ *for client group $D$ satisfies $\epsilon_g$-global group fairness if for all $y \in \mathcal{Y}, a \in \mathcal{A}$*

$$\left| \Pr_D \left\{ \widetilde{Y}(X, A, C) = y \,|\, Y = y, A = a \right\} - \Pr_D \left\{ \widetilde{Y}(X, A, C) = y \,|\, Y = y \right\} \right| \leq \epsilon_g \tag{1}$$

*(2). $\epsilon_l$-**Local Group Fairness:** The outcome predictor* $\widetilde{Y} : \mathcal{X} \times \mathcal{A} \times \mathcal{C} \to \mathcal{Y}$ *satisfies $\epsilon_l$-local group fairness if for all $c \in \mathcal{C}, y \in \mathcal{Y}, a \in \mathcal{A}$*

$$\left| \Pr_D \left\{ \widetilde{Y}(X, A, C) = y \,|\, Y = y, A = a, C = c \right\} - \Pr_D \left\{ \widetilde{Y}(X, A, C) = y \,|\, Y = y, C = c \right\} \right| \leq \epsilon_l \tag{2}$$

*(3). $\epsilon_c$-**Client Fairness:** The outcome predictor* $\widetilde{Y} : \mathcal{X} \times \mathcal{A} \times \mathcal{C} \to \mathcal{Y}$ *satisfies $\epsilon_c$-client fairness if for all $c \in \mathcal{C}$*

$$\left| \Pr_D \left\{ \widetilde{Y}(X, A, C) = Y \,|\, C = c \right\} - \Pr_D \left\{ \widetilde{Y}(X, A, C) = Y \right\} \right| \leq \epsilon_c \tag{3}$$

This section defines global and local group fairness w.r.t multi-class *Equal Opportunity* (Hardt et al., 2016) and client fairness w.r.t *Disparate Mistreatment* (Zafar et al., 2017a). Our framework extends to other fairness metrics (AppendixC). Practitioners can choose metrics based on application needs.

This paper considers selecting an outcome predictor $\widetilde{Y}$ that satisfies Eq. (1), (2), (3) while maximizing the accuracy. Consider the true positive of $\widetilde{Y}$ for class $y$, group $a$ and client $c$: $\mathrm{TP}_{ac}^y(\widetilde{Y}) = \mathbb{E}_{\mathrm{Pr}_{X|Y,A,C}}\left[\mathbb{1}(\widetilde{Y}(X,a,c) = y)\right]$. The predictor's accuracy is a linear combination of true positives:

$$\mathrm{acc}(\widetilde{Y}) = \sum_{y\in\mathcal{Y}}\sum_{c\in\mathcal{C}}\sum_{a\in\mathcal{A}}\left[\mathrm{Pr}_D(Y = y, A = a, C = c)\cdot\mathrm{TP}_{ac}^y(\widetilde{Y})\right]$$

The following section defines the feasible set of true positives for multi-class problems.

## 2.2 Region Under ROC Surface

This section defines Receiver Operating Characteristic (ROC) surface and formally prove that region under ROC surfaces gives feasible set of true positives.

**Definition 2.2** *Let the function $R : \mathcal{X} \times \mathcal{A} \times \mathcal{C} \to [0,1]^N$ have components $R(x,a,c) = [r_1(x,a,c), r_2(x,a,c), \cdots r_N(x,a,c)]$. $R$ is a Bayesian Optimal Score Function if for all $y \in \mathcal{Y}$, its components have: $r_y(x,a,c) = \mathrm{Pr}_D(Y = y|X = x, A = a, C = c)$*

The Bayesian Optimal Score function maps inputs to a probability distribution over $\mathcal{Y}$, with each element representing the probability that an individual $(x,a,c)$ belongs to each class. For simplicity, we drop the input arguments and write $R(X,A,C)$ or $\widetilde{Y}(X,A,C)$ as $R$ or $\widetilde{Y}$.

**Definition 2.3** *Let $R : \mathcal{X} \times \mathcal{A} \times \mathcal{C} \to [0,1]^N$ be the Bayesian Optimal Score Function and $\boldsymbol{\theta} = [\theta_1, \theta_2, \cdots, \theta_N] \in [0,1]^N$ be a given vector. The outcome predictor $\widetilde{Y}_{\boldsymbol{\theta}} : \mathcal{X} \times \mathcal{A} \times \mathcal{C} \to \mathcal{Y}$ that takes value of: $\widetilde{Y}_{\boldsymbol{\theta}} = y$, if, $\theta_y r_y(x,a,c) = \max_{i=1}^N \theta_i r_i(x,a,c)$ is derived from the score function $R$ via $\boldsymbol{\theta}$. The set of all derived outcome predictors is $\{\widetilde{Y}_{\boldsymbol{\theta}}\}_{\boldsymbol{\theta}\in\mathbb{R}_{\geq 0}^N}$.*

A derived outcome predictor returns the highest value of the score weighted by $\boldsymbol{\theta}$. It generalizes the threshold test predictor from Hardt et al. (2016) used in binary classification.

Consider the vector in $[0,1]^N$ that represents the true positives for client $c$ and group $a$ of the *derived outcome predictor* for a given $\boldsymbol{\theta}$. We denote the vector of true positives of the predictor $\widetilde{Y}_{\boldsymbol{\theta}}$ as $\mathbf{TP}_{ac}(\widetilde{Y}_{\boldsymbol{\theta}}) := [\mathrm{TP}_{ac}^1(\widetilde{Y}_{\boldsymbol{\theta}}), \ldots, \mathrm{TP}_{ac}^N(\widetilde{Y}_{\boldsymbol{\theta}})]^T$. The ROC surface for the multi-class setting is composed of the true positive vectors of all derived outcome predictors, $\{\widetilde{Y}_{\boldsymbol{\theta}}\}_{\boldsymbol{\theta}\in\mathbb{R}_{\geq 0}^N}$. The ROC surface for client $c$ and group $a$ is thus defined as:

$$\mathrm{ROC}_{ac} := \{\mathbf{TP}_{ac}(\widetilde{Y}_{\boldsymbol{\theta}}) : \forall \boldsymbol{\theta} \in \mathbb{R}_{\geq 0}^N\} \tag{4}$$

The ROC surfaces differ across clients as the data distributions vary across different clients in the FL system. We consider the region under the ROC surface (RUS). The RUS is defined with respect to the separating hyperplanes that distinguishes the region under the ROC surface from other regions.

**Definition 2.4** *Let $\boldsymbol{TP}_{ac}(\widetilde{Y}_{\boldsymbol{\theta}})$ be the point representing the true positive of the derived outcome predictor for a given $\boldsymbol{\theta} = [\theta_1, \cdots, \theta_N]$, the hyperplane of $\boldsymbol{TP}_{ac}(\widetilde{Y}_{\boldsymbol{\theta}})$ is defined as:*

$$\left\{\mathbf{x} \in \mathbb{R}^N | \mathbf{v}_{\boldsymbol{\theta}}^T\mathbf{x} = \mathbf{v}_{\boldsymbol{\theta}}^T\boldsymbol{TP}_{ac}(\widetilde{Y}_{\boldsymbol{\theta}})\right\}$$

*where,*

$$\mathbf{v}_{\boldsymbol{\theta}}^T = [\theta_y\mathrm{Pr}_D(Y = y|A = a, C = c), \forall y \in \mathcal{Y}]$$

For any given $\boldsymbol{\theta} \in \mathbb{R}^N \geq \mathbf{0}$, it has a separating hyperplane such that the set $\{\mathbf{x} \in \mathbb{R}^N | \mathbf{v}_{\boldsymbol{\theta}}^T\mathbf{x} > \mathbf{v}_{\boldsymbol{\theta}}^T\boldsymbol{TP}_{ac}(\widetilde{Y}_{\boldsymbol{\theta}})\}$ excludes the RUS.

**Definition 2.5** *The region under $\mathrm{ROC}_{ac}$ is: $D_{ac} = \bigcap_{\boldsymbol{\theta}\in\mathbb{R}_{\geq 0}^N}\left\{\mathbf{x} \in [0,1]^N | \mathbf{v}_{\boldsymbol{\theta}}^T\mathbf{x} \leq \mathbf{v}_{\boldsymbol{\theta}}^T\boldsymbol{TP}_{ac}(\widetilde{Y}_{\boldsymbol{\theta}})\right\}$*

**Proposition 2.6** *(Appendix G.1) Let $D_{ac}$ be the region defined in Def.2.5. Then, $D_{ac}$ is a convex set. For any predictor $\widetilde{Y} : \mathcal{X} \times \mathcal{A} \times \mathcal{C} \to \mathcal{Y}$, let the point representing true positives of $\widetilde{Y}$ be:* $\boldsymbol{TP}_{ac}(\widetilde{Y}) = [\mathrm{TP}^1_{ac}(\widetilde{Y}), \cdots, \mathrm{TP}^N_{ac}(\widetilde{Y})]$. *Then, $\boldsymbol{TP}_{ac}(\widetilde{Y})$ lies in $D_{ac}$.*

Proposition 2.6 asserts that for group $a$ in client $c$, any outcome predictor that is a map $\widetilde{Y} : \mathcal{X} \times \mathcal{A} \times \mathcal{C} \to \mathcal{Y}$, has its true positives lying in the convex set $D_{ac}$. Thus, $D_{ac}$ is the convex set representing the feasible region of true positives.

## 3 BALANCING ACCURACY AND DISTRIBUTIVE FAIRNESS CONCEPTS IN FL

This section provides a framework that balances fairness concepts and model performance in FL.

Consider the loss function $\ell : \mathcal{Y} \times \mathcal{Y} \to \{0, 1\}$ that takes values: $\ell(\widetilde{y}, y) = \mathbb{1}(\widetilde{y} \neq y)$ for any $\widetilde{y}, y \in \mathcal{Y}$, where $\mathbb{1}(\cdot)$ is the indicator function. Any predictor $\widetilde{Y} : \mathcal{X} \times \mathcal{A} \times \mathcal{C} \to \mathcal{Y}$ that satisfies the following optimization problem is an $\epsilon$-**fair optimal outcome predictor**

$$
\begin{aligned}
&\text{minimize} && \mathbb{E}_D\left[\ell(\widetilde{Y}(X, A, C), Y)\right] \\
&\text{with respect to} && \widetilde{Y} : \mathcal{X} \times \mathcal{A} \times \mathcal{C} \to \mathcal{Y} \\
&\text{subject to} && \text{eq.}(1), (2) \text{ and eq.}(3)
\end{aligned}
\tag{5}
$$

Optimization (5) can be recast as the following convex program. Let $z^y_{ac}$ be the variables for the outcome predictor $\widetilde{Y}$,

$$
z^y_{ac} = \Pr_D(\widetilde{Y} = y | Y = y, A = a, C = c)
$$

that represents the true positives of $\widetilde{Y}$ for class $y$, group $a$ and client $c$. Proposition 3.1 asserts that if $\{z^y_{ac}\}_{\mathcal{Y}, \mathcal{A}, \mathcal{C}}$ satisfies the following convex program, then $\widetilde{Y}$ is a $\epsilon$-fair optimal outcome predictor.

**Proposition 3.1** *(Appendix G.2) Let the vector $\mathbf{z} \in \mathbb{R}^{2NK}$:*

$$
\mathbf{z}^T = \begin{bmatrix} \mathbf{z}^T_{01} & \mathbf{z}^T_{11} & \mathbf{z}^T_{02} & \mathbf{z}^T_{12} \cdots & \mathbf{z}^T_{0K} & \mathbf{z}^T_{1K} \end{bmatrix},
$$

$$
\text{with,} \quad \mathbf{z}^T_{ac} = \begin{bmatrix} z^1_{ac} & z^2_{ac} & z^3_{ac} & \cdots & z^N_{ac} \end{bmatrix} \in \mathbb{R}^N
$$

*satisfy the following convex program*

$$
\begin{aligned}
&\text{minimize:} && \mathbf{c}^T \mathbf{z} \\
&\text{with respect to:} && \mathbf{z} \in \mathbb{R}^{2NK} \\
&\text{subject to:} && -\mathbf{b} \leq \mathbf{A}\mathbf{z} \leq \mathbf{b} \\
&&& \mathbf{z}_{ac} \in D_{ac}, \forall a \in \mathcal{A}, c \in \mathcal{C}
\end{aligned}
\tag{6}
$$

*then, the outcome predictor $\widetilde{Y} : \mathcal{X} \times \mathcal{A} \times \mathcal{C} \to \mathcal{Y}$ that satisfies eq. (7) for all $y \in \mathcal{Y}, a \in \mathcal{A}, c \in \mathcal{C}$*

$$
\Pr(\widetilde{Y} = y | Y = y, A = a, C = c) = z^y_{ac}
\tag{7}
$$

*is a $\epsilon$-fair optimal outcome predictor. The optimal accuracy for a $\epsilon$-fair outcome predictor is $-\mathbf{c}^T \mathbf{z}$.*

The parameters of (6), $\mathbf{A} \in \mathbb{R}^{(N+NK+K) \times 2NK}$, $\mathbf{c} \in \mathbb{R}^{2NK}$ and $\mathbf{b} \in \mathbb{R}^{N+NK+K}$ are detailed in Appendix B. The first $N$ inequalities represent global group fairness, the next $NK$ inequalities represent local group fairness and the last $K$ represents client fairness. $\mathbf{b}$ specifies fairness level. $D_{ac}$ is the region under the ROC defined in Def. 2.5. Optimization (6) is a convex program since $D_{ac}$ is a convex set. We can enforce fairness w.r.t other metric by replacing the parameter of the convex program (detailed in Appendix C). Appendix G.4 shows that problem (6) always has solutions.

Since $D_{ac}$ is the feasible region for the true positives of all outcome predictors (Prop. 2.6), the predictor in Proposition 3.1 is fair and has the optimal accuracy. The convex program (6) thus presents the *inherent trade-off* between global, local group fairness, client fairness and accuracy.

The computational complexity of generating the convex set $D_{ac}$ increases exponentially w.r.t. the number of classes (Landgrebe & Duin, 2008). Instead of solving the convex program (6) directly, this paper uses a convex polytope $\widehat{D_{ac}}$ as an inner approximation of $D_{ac}$. The vertices of the polytope are

points on $D_{ac}$. The approach for estimating $D_{ac}$ is detailed in Appendix H. This approach allows us to reformulate the problem (6) as a linear program (LP).

The first approximation of $D_{ac}$ is the simplex $\widehat{D_{ac}}$ whose vertices are standard basis vectors $\{\mathbf{e}_y\}_{y\in\mathcal{Y}}$ and the true positives of the derived outcome predictor (Def. 2.3), $\mathbf{TP}_{ac}(\widetilde{Y}_{\boldsymbol{\theta}_1})$, where $\boldsymbol{\theta}_1 = \frac{1}{N}\mathbf{1}_N$,

$$\widehat{D_{ac}} = \{f_0\mathbf{TP}_{ac}(\widetilde{Y}_{\boldsymbol{\theta}_1}) + \sum_{y=1}^{N} f_y\mathbf{e}_y \mid \sum_{i=0}^{N} f_i = 1 \text{ and } \forall i, f_i \geq 0\}$$

$\widehat{D_{ac}}$ can be represented using $(N+1)$ inequalities. Any point $\mathbf{v} \in \mathbb{R}^N$ that lies in $\widehat{D_{ac}}$ must satisfy:
$$\mathbf{K}_{ac}\mathbf{v} \leq \mathbf{l}_{ac}$$
where $\mathbf{K}_{ac} \in \mathbb{R}^{(N+1)\times N}$ and $\mathbf{l}_{ac} \in \mathbb{R}^{(N+1)}$ are detailed in Appendix E. We reformulate the problem (6) as an LP:

$$
\begin{array}{ll}
\text{minimize:} & \mathbf{c}^T\mathbf{z} \\
\text{with respect to:} & \mathbf{z} \in \mathbb{R}^{2NK} \\
\text{subject to:} & -\mathbf{b} \leq \mathbf{A}\mathbf{z} \leq \mathbf{b} \\
& \mathbf{K}_{ac}\mathbf{z}_{ac} \leq \mathbf{l}_{ac}, \forall a \in \mathcal{A}, c \in \mathcal{C}
\end{array}
\tag{8}
$$

The LP (8) identifies the true positives for the fair outcome predictor. The next step is to construct a classifier that satisfies those true positives. The following proposition establishes the existence and uniqueness of such a fair outcome predictor given the LP solution.

**Proposition 3.2** *(Appendix G.3) Let* $\mathbf{z} \in \mathbb{R}^{2NK}$ *be the solution of the LP (8)*
$$\mathbf{z}^T = \begin{bmatrix} \mathbf{z}_{01}^T & \mathbf{z}_{11}^T & \mathbf{z}_{02}^T & \mathbf{z}_{12}^T \cdots & \mathbf{z}_{0K}^T & \mathbf{z}_{1K}^T \end{bmatrix}$$
*and* $\mathrm{TP}_{ac}^y(\widetilde{Y}_{\boldsymbol{\theta}_1})$ *be the true positive of the* derived outcome predictor *by* $\boldsymbol{\theta}_1$ *For all* $a \in \mathcal{A}, c \in \mathcal{C}$, *let* $\boldsymbol{\beta}_{ac} = [\beta_{ac}^0, \beta_{ac}^1, \cdots, \beta_{ac}^N]$ *be the solution of the following linear algebraic equation (LAE),*
$$\mathbf{G}_{ac}\boldsymbol{\beta}_{ac} = \boldsymbol{\gamma}_{ac} \tag{9}$$
*where, the parameter* $\mathbf{G}_{ac} \in \mathbb{R}^{(N+1)\times(N+1)}, \boldsymbol{\gamma}_{ac} \in \mathbb{R}^{N+1}$, *are detailed in Appendix F. Then. the predictor* $\widetilde{Y}_{\boldsymbol{\beta}_{ac}}$ *that takes value,*
$$
\widetilde{Y}_{\boldsymbol{\beta}_{ac}}(x,a,c) = \begin{cases} \widetilde{Y}_{\boldsymbol{\theta}_1}(x,a,c), & \text{with the probability } \beta_{ac}^0 \\ y, & \text{with the probability } \beta_{ac}^y, \forall y \in \mathcal{Y} \end{cases}
\tag{10}
$$
*is a fair outcome predictor. There always exists a unique set of parameters* $\{\boldsymbol{\beta}_{ac}\}_{\mathcal{A},\mathcal{C}}$, *where* $\boldsymbol{\beta}_{ac} \in [0,1]^{N+1}$ *and* $|\boldsymbol{\beta}_{ac}|_{\ell_1} = 1$ *that satisfies the LAE.*

Proposition 3.2 gives the fair outcome predictor from the LP solution. Combining the LP (8) and LAE (9), our framework first solves the LP (8) that identifies the true positives for the fair outcome predictor. Then, the fair outcome predictor (28) can be uniquely determined by solving the LAE (9).

# 4 TRAINING FAIR OUTCOME PREDICTORS IN FL

This section presents how LP (8) and the LAE (9) are integrated into a FL system to construct a fair outcome predictor. The training procedure is outlined below:

(1). **Train the Bayesian Optimal Score Function via `FedAvg`**: Clients and server collaboratively train the Bayesian Optimal Score Function $R : \mathcal{X} \times \mathcal{A} \times \mathcal{C} \rightarrow [0,1]^N$ by minimizing the loss function: $\mathbb{E}_D[\mathbb{1}(Y=y)\log r_y(X,A,C)]$ using `FedAvg` algorithm (McMahan et al., 2017), which gives the empirical estimation of the score function $R$.

(2). **Local Prediction and Statistics Calculation**: Each client generates the outcome $\widetilde{Y}_{\boldsymbol{\theta}_1}(X,A,C)$ and computes the following statistics $\forall y \in \mathcal{Y}, a \in \mathcal{A}$. Then clients send these statistics to the server.

$$
\begin{aligned}
\mathrm{Pr}_D(\widetilde{Y}_{\boldsymbol{\theta}_1} = y, Y = y, A = a \mid C = c) &= \frac{\#(\widetilde{Y}_{\boldsymbol{\theta}_1} = y, Y = y, A = a)}{\# \text{ samples in client } c} \\
\mathrm{Pr}_D(Y = y, A = a \mid C = c) &= \frac{\#(Y = y, A = a)}{\# \text{ samples in client } c}
\end{aligned}
\tag{11}
$$

---

**Algorithm 1** Fair Outcome Predictor

---

**Input:** The outcome predictor: $\widetilde{Y}_{\boldsymbol{\theta}_1} : \mathcal{X} \times \mathcal{A} \times \mathcal{C} \to \mathcal{Y}$, the client $c$'s $\boldsymbol{\beta}_{ac}, \forall a \in \mathcal{A}$
$\boldsymbol{\beta}_{ac}^T = [\beta_{ac}^0, \beta_{ac}^1, \beta_{ac}^2, \cdots \beta_{ac}^N] \in [0, 1]^{N+1}$
**Output:** Fair outcome predictor $\widetilde{Y}_{\boldsymbol{\beta}_{ac}} : \mathcal{X} \times \mathcal{A} \times \mathcal{C} \to \mathcal{Y}$
   1. randomly sample $s \sim U(0, 1)$, the uniform distribution between [0,1]
   2. Construct $\widetilde{Y}_{\boldsymbol{\beta}_{ac}}(x, a, c)$ as

$$\widetilde{Y}_{\boldsymbol{\beta}_{ac}}(x, a, c) = \begin{cases} \widetilde{Y}_{\boldsymbol{\theta}_1}(x, a, c), \text{ if } s \leq \beta_{ac}^0 \\ 1, \text{ if } \beta_{ac}^0 < s \leq \sum_{i=0}^{i=1} \beta_{ac}^i \\ 2, \text{ if } \sum_{i=0}^{i=1} \beta_{ac}^i < s \leq \sum_{i=0}^{i=2} \beta_{ac}^i \\ \cdots \\ N, \text{ if } \sum_{i=0}^{i=N-1} \beta_{ac}^i < s \leq \sum_{i=0}^{i=N} \beta_{ac}^i \end{cases}$$

**return** $\widetilde{Y}_{\boldsymbol{\beta}_{ac}}$

---

The private profile $x \in \mathcal{X}$ are kept locally in this step. Differential Privacy mechanisms (Dwork, 2006) can be added in those statistics. Details of Laplacian mechanism are in Appendix I.5.

(3). **Solve the LP**: The server construct LP (8) using the statistics sent by the clients. The parameters of the LP is detailed in Appendix D. The server finds the minimizer of the LP $\mathbf{z}^T = \begin{bmatrix} \mathbf{z}_{01}^T & \mathbf{z}_{11}^T & \mathbf{z}_{02}^T & \mathbf{z}_{12}^T \cdots & \mathbf{z}_{0K}^T & \mathbf{z}_{1K}^T \end{bmatrix}$ and sends the corresponding minimizer $\mathbf{z}_{0c}^T, \mathbf{z}_{1c}^T$ to client $c$, where $c = 1, 2, \cdots, K$.

(4). **Solve the LAE and Make the Fair Prediction**: Each client constructs the LAE (9) using the $\mathbf{z}_{0c}^T, \mathbf{z}_{1c}^T$ sent from the clients and solve the LAE. The solution of the LAE $\boldsymbol{\beta}_{ac}$ is used in Algorithm 1 to make fair predictions.

LP and LAE can be solved in polynomial time complexity w.r.t. the number of variables and constraints (Karmarkar, 1984). Our algorithm is scalable for large scale distributed systems.

## 5 EXPERIMENTS

We conduct experiments on three datasets. Results show our framework is effective for enforcing different fairness concepts. Compared with SOTA, our framework achieves competitive accuracy for enforcing fairness while having smaller communicational and computational cost.

**Dataset:** The datasets we used are: *Adult*, *ACSPublicCoverage* and *HM10000*. For each dataset, we pre-specify a fairness metric for local, global, client fairness and solve the corresponding LP.

(1) **Adult** (Asuncion & Newman, 2007) predicts whether an individual earns over 50K/year. The data is split into two clients based on PhD status; gender is the sensitive attribute. Global and local fairness use Statistical Parity. Client fairness uses Disparate Mistreatment.

(2) **ACSPublicCoverage** (Ding et al., 2021) predicts eligibility for public health insurance using data from 50 U.S. states. The sensitive attribute is race (white/non-white). Global and client fairness use Equal Opportunity (EOp). Client fairness uses Disparate Mistreatment as fairness metric.

(3) **HM10000** (Tschandl et al., 2018) is a dermatoscopic image dataset for 4-class diagnosis. We split it into five clients with varied sensitive attribute (gender) makeup. All fairness metrics are multi-class Equal Opportunity, which generalizes Equalized Odds in Hardt et al. (2016) for binary tasks.

**Baselines:** We compare our framework with baselines that enforce: (1) Global fairness: `FairFed` (Ezzeldin et al., 2023), `Fair-FATE` (Salazar et al., 2023) (2) Local fairness: FCFL Cui et al. (2021) (3) Client fairness: `Agnostic-FL(AFL)` (Mohri et al., 2019), `q-FFL` (Li et al., 2019). FCFL (Cui et al., 2021) addresses both local and client fairness and `EquiFL` (Makhija et al., 2024) `LOGO` (Zhang et al., 2025) addresses both local and global fairness. Since most baselines only support binary-class settings, we compare against them on binary tasks.

**Evaluation:** We assess the model from four perspective: (1) Model performance is measured by accuracy over all samples. For different fairness concepts, we use different metrics (Statistical

Parity (SP), Equal Opportunity (EOp), Equalized Odds (EO) and Disparate Mistreatment (DM)) and measure it accordingly. We use SP Difference $\Delta_{SP}$ (difference in positive rate) for SP, EOp Difference $\Delta_{EOp}$ (difference in true positive rate) for EOp, and EO difference $\Delta_{EO}$ (maximum difference in true positives across all classes) for multi-class Equal Opportunity. Disparate Mistreatment is measured by Accuracy Disparity $\Delta_{DM}$. (2) Local fairness is measured by average disparity within each client. (3) Global fairness, measured by disparity across the entire dataset. (4) Client fairness, measured by the disparity between the client. The detailed descriptions of dataset, baselines, evaluation and training details are in Appendix I.1. Our main experimental results as follows:

Table 1: Accuracy, local, global group disparity and client disparity of all algorithms for binary tasks. ($\cdot$) indicates the fairness concepts each method addresses: (g) Global, (l) Local, (c) Client

| Framework | Adult (gender) | | | | PublicCoverage (race) | | | |
|---|---|---|---|---|---|---|---|---|
| | $\Delta_{SP}^{local}$ ($\downarrow$) | $\Delta_{SP}^{global}$ ($\downarrow$) | $\Delta_{DM}^{client}$ ($\downarrow$) | Acc ($\uparrow$) | $\Delta_{EOp}^{local}$ ($\downarrow$) | $\Delta_{EOp}^{global}$ ($\downarrow$) | $\Delta_{DM}^{client}$ ($\downarrow$) | Acc ($\uparrow$) |
| FedAvg | $0.29 \pm 0.04$ | $0.23 \pm 0.03$ | $0.10 \pm 0.04$ | $\mathbf{84.50 \pm 0.60}$ | $0.15 \pm 0.04$ | $0.09 \pm 0.03$ | $0.27 \pm 0.05$ | $\mathbf{77.80 \pm 0.40}$ |
| FairFed (g) | $0.07 \pm 0.03$ | $0.08 \pm 0.01$ | $0.36 \pm 0.08$ | $81.20 \pm 0.70$ | $0.09 \pm 0.03$ | $0.01 \pm 0.01$ | $0.25 \pm 0.08$ | $76.40 \pm 0.40$ |
| Fair-FATE (g) | $0.11 \pm 0.07$ | $0.03 \pm 0.01$ | $0.11 \pm 0.50$ | $80.40 \pm 0.30$ | $0.08 \pm 0.05$ | $0.02 \pm 0.01$ | $0.40 \pm 0.04$ | $75.90 \pm 0.70$ |
| EquiFL (g & l) | $0.05 \pm 0.04$ | $0.06 \pm 0.04$ | $0.26 \pm 0.05$ | $77.00 \pm 2.70$ | $0.05 \pm 0.01$ | $0.04 \pm 0.01$ | $0.39 \pm 0.03$ | $76.40 \pm 0.20$ |
| LOGO (g & l) | $0.05 \pm 0.01$ | $0.03 \pm 0.01$ | $0.09 \pm 0.02$ | $81.50 \pm 0.04$ | $0.05 \pm 0.01$ | $0.02 \pm 0.01$ | $0.33 \pm 0.03$ | $76.30 \pm 0.30$ |
| Agnostic-FL (c) | $0.27 \pm 0.03$ | $0.33 \pm 0.03$ | $0.08 \pm 0.01$ | $80.10 \pm 0.40$ | $0.15 \pm 0.04$ | $0.18 \pm 0.02$ | $0.15 \pm 0.02$ | $76.00 \pm 0.80$ |
| q-FFL (c) | $0.27 \pm 0.04$ | $0.36 \pm 0.05$ | $0.07 \pm 0.02$ | $81.80 \pm 0.60$ | $0.20 \pm 0.03$ | $0.09 \pm 0.03$ | $0.14 \pm 0.02$ | $75.90 \pm 0.10$ |
| FCFL (l & c) | $0.05 \pm 0.01$ | $0.05 \pm 0.02$ | $0.06 \pm 0.04$ | $81.60 \pm 0.40$ | $0.05 \pm 0.02$ | $0.03 \pm 0.02$ | $0.16 \pm 0.05$ | $76.50 \pm 0.40$ |
| *Ours (all)* | $\mathbf{0.04 \pm 0.01}$ | $0.01 \pm 0.01$ | $\mathbf{0.02 \pm 0.01}$ | $78.70 \pm 0.20$ | $\mathbf{0.04 \pm 0.01}$ | $0.01 \pm 0.01$ | $\mathbf{0.08 \pm 0.04}$ | $70.90 \pm 0.20$ |
| *Ours (global)* | $0.15 \pm 0.01$ | $\mathbf{0.01 \pm 0.01}$ | $0.04 \pm 0.01$ | $81.00 \pm 0.30$ | $0.16 \pm 0.01$ | $\mathbf{0.01 \pm 0.00}$ | $0.49 \pm 0.04$ | $76.60 \pm 0.10$ |
| *Ours (local)* | $0.04 \pm 0.02$ | $0.02 \pm 0.01$ | $0.08 \pm 0.02$ | $81.00 \pm 0.30$ | $0.05 \pm 0.02$ | $0.02 \pm 0.01$ | $0.26 \pm 0.04$ | $76.60 \pm 0.40$ |
| *Ours (client)* | $0.39 \pm 0.03$ | $0.35 \pm 0.01$ | $0.08 \pm 0.01$ | $81.20 \pm 0.10$ | $0.34 \pm 0.02$ | $0.24 \pm 0.01$ | $0.16 \pm 0.01$ | $75.70 \pm 0.30$ |

Table 2: Accuracy, local, global group disparity and client disparity for multi-class tasks

| Method | HM10000 (s1) | | | | HM10000 (s5) | | | |
|---|---|---|---|---|---|---|---|---|
| | $\Delta_{EO}^{local}$ ($\downarrow$) | $\Delta_{EO}^{global}$ ($\downarrow$) | $\Delta_{EO}^{client}$ ($\downarrow$) | Acc ($\uparrow$) | $\Delta_{EO}^{local}$ ($\downarrow$) | $\Delta_{EO}^{global}$ ($\downarrow$) | $\Delta_{EO}^{client}$ ($\downarrow$) | Acc ($\uparrow$) |
| FedAvg | $0.28 \pm 0.04$ | $0.26 \pm 0.02$ | $0.01 \pm 0.02$ | $\mathbf{81.1 \pm 0.7}$ | $0.44 \pm 0.03$ | $0.31 \pm 0.01$ | $0.09 \pm 0.01$ | $\mathbf{82.1 \pm 0.8}$ |
| *Ours (all)* | $\mathbf{0.03 \pm 0.02}$ | $0.03 \pm 0.01$ | $0.02 \pm 0.00$ | $70.6 \pm 0.8$ | $\mathbf{0.03 \pm 0.01}$ | $\mathbf{0.02 \pm 0.01}$ | $\mathbf{0.01 \pm 0.00}$ | $67.2 \pm 0.6$ |
| *Ours (global)* | $0.42 \pm 0.03$ | $\mathbf{0.02 \pm 0.01}$ | $0.20 \pm 0.01$ | $77.8 \pm 1.8$ | $0.50 \pm 0.05$ | $0.03 \pm 0.07$ | $0.09 \pm 0.01$ | $77.4 \pm 0.1$ |
| *Ours (local)* | $0.07 \pm 0.04$ | $0.05 \pm 0.01$ | $0.21 \pm 0.01$ | $70.7 \pm 0.4$ | $0.12 \pm 0.01$ | $0.23 \pm 0.01$ | $0.22 \pm 0.00$ | $67.4 \pm 1.3$ |
| *Ours (client)* | $0.35 \pm 0.01$ | $0.20 \pm 0.01$ | $\mathbf{0.01 \pm 0.01}$ | $80.8 \pm 0.1$ | $0.41 \pm 0.07$ | $0.06 \pm 0.01$ | $0.01 \pm 0.01$ | $71.4 \pm 0.3$ |

**Enforcing Fairness in FL:** We adjust our framework to enforce either all fairness concepts (with all fairness constraints in the LP) or a single one (with just one fairness constraint in the LP). We adjust $(\epsilon_g, \epsilon_l, \epsilon_c)$ so the fairness level is similar to the baselines, we then compare the accuracy under comparable fairness levels. Table 1 reports local, global group, client disparity and accuracy of our framework and baselines. Compared to FedAvg, our framework (Ours (all)) reduces local disparity by 86% ($0.29 \rightarrow 0.04$) on *Adult* and 73% ($0.15 \rightarrow 0.04$) on *PublicCoverage*; global disparity by 91% ($0.23 \rightarrow 0.01$) for *Adult* and 88% ($0.09 \rightarrow 0.01$) for *PublicCoverage*, client disparity by 80% ($0.10 \rightarrow 0.02$) for *Adult* and 70% ($0.27 \rightarrow 0.08$) for *PublicCoverage*. These results show ours effectively enforce different fairness concepts in FL. The Pareto frontiers illustrating the trade-offs between accuracy and each D-fairness concept are in Appendix I.2. Those results demonstrate by changing the parameters in LP (8), the fairness levels can be flexibility adjusted. Our framework is designed to balance different fairness concepts, however, when enforcing a single fairness objective, Table 1 shows that our framework maintains better or competitive accuracy across all baselines.

**Communication and Computational Costs:** Table 3 reports the number of communication rounds and computational cost. Communication rounds reflect communication efficiency and convergence time measures computational cost. The number of communication rounds for our method is 33% smaller for the *Adult* and *PublicCoverage* compared to in-processing baselines that has smallest communication round (Fair-FATE, FairFed). The computational time is 33% less than for *Adult* and 53% less for *PubCoverage* than the best-performing baseline (FairFed). Our framework enforces fairness with smaller communication and computational costs.

**Fairness under Different Data Heterogeneity:** This section examines how data heterogeneity impacts three D-Fair concepts. Data heterogeneity in FL refers to the scenarios that data across clients in FL are non-i.i.d. Experiments were conducted on the multi-class HM 10000 dataset, where the client makeup w.r.t the sensitive attribute was varied. In Scenario 1 (s1), data is i.i.d.

Table 3: The number of communication round and time for convergence for all in-processing methods

| Frameworks | Adult (2 clients) | | | | | PublicCoverage (50 clients) | | | | |
|---|---|---|---|---|---|---|---|---|---|---|
| | FairFed | Fair-FATE | EquiFL | AFL, q-FFL, FCFL | Ours | FairFed | Fair-FATE | EquiFL | AFL, q-FFL, FCFL | Ours |
| # of communication rounds ≈ | 15 | 15 | 30 | > 100 | **10** | 15 | 15 | 30 | >100 | 10 |
| Total time for convergence (s) ≈ | 30 | 41 | 54 | > 220 | **20** | 484 | 564 | 852 | >1000 | **228** |

across clients with sensitive proportions $[0.4, 0.4, 0.4, 0.4, 0.4]$. For s2-s5, the distribution becomes increasingly skewed: $[0.2, 0.3, 0.4, 0.5, 0.6]$, $[0.3, 0.3, 0.3, 0.7, 0.7]$, $[0.1, 0.1, 0.1, 0.9, 0.9]$, and $[0.05, 0.05, 0.05, 0.95, 0.95]$. Results of s1 and s5 are in Table 2; s2–s4 are in Appendix I.3.

Table 2 shows the accuracy cost of enforcing global fairness is 4% ($81.1 \rightarrow 77.8$) for s1 and 5.7% ($82.1 \rightarrow 77.4$) for s5. For local fairness, the cost is 12.8% ($81.1 \rightarrow 70.7$) for s1 and 17.9% ($82.1 \rightarrow 67.4$) for s5. For client fairness, the cost is 0.4% ($81.1 \rightarrow 80.8$) for s1 and 13% ($82.1 \rightarrow 71.4$) for s5. Under high heterogeneity (s5), the accuracy cost of global fairness is similar to s1, but local and client fairness incur higher costs. When the data is i.i.d., enforcing local fairness leads to improved global fairness. However, enforcing global fairness does not necessarily imply local fairness. This is because our framework uses client-specific predictor, a zero aggregated disparity does not guarantee zero disparity at the individual client. When sensitive attributes are unevenly distributed, enforcing client fairness reduces global disparity by 41% reduction in global disparity ($0.31 \rightarrow 0.18$).

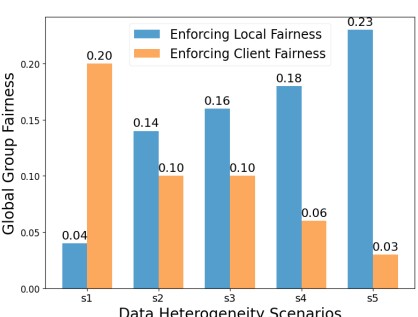

Figure 2: Global Group Disparity when enforcing local & client fairness under different data heterogeneity

We further investigate how local group and client fairness impact global group fairness under data heterogeneity. Results in Fig. 2 show how group fairness disparity changes as data heterogeneity increases when applying either local or client fairness constraints. When local group fairness is enforced, moving from s1 to s5, i.e., as the distribution of sensitive attributes becomes more skewed across clients, the global group disparity increases. Together with the results in Table 2, we conclude that enforcing local fairness improves global fairness when data is i.i.d. across clients, but this improvement on global fairness diminishes as the sensitive attribute becomes more imbalanced across clients. Similarly, enforcing client fairness enhances global group fairness when the sensitive attribute is completely skewed across clients, but this improvement declines as the sensitive attribute becomes more independent of client identity.

**Relaxation of the Convex Program:** This paper uses the simplex $\widehat{D_{ac}}$ as an inner approximation of the RUS. This section empirically investigates how this relaxation affects the accuracy of fair predictors. Results in Appendix I.4 compare the linear program (LP) relaxation with the convex program (CP). Results show that the LP closely approximates the optimal accuracy under fairness constraints. While the CP yields slightly better accuracy, its computational cost increase exponentially w.r.t the number of class as the computational complexity of generating the convex set $D_{ac}$ is exponential w.r.t the number of class (Landgrebe & Duin, 2008). The LP has polynomial complexity in the number class, making it scalable for large FL systems.

**Privacy Protection of Local Statistics:** Local differential privacy (DP) can be applied during the communication of local statistics in step (2) of our training pipeline in Sec 4. to protect client-level privacy. Appendix I.5 illustrates how DP mechanisms affect the fairness and accuracy of our algorithm. Our framework is effective in enforcing all fairness concepts under a 0.01-differentially private setting. As $\epsilon$ decreases (i.e., privacy protection becomes stronger), local, global and client disparities tend to increase, which shows the trade-off between privacy and fairness under our framework.

## 6 CONCLUSIONS

This paper discusses motivating applications where multiple D-Fair concepts are required simultaneously in FL. We thus introduce a post-processing framework that enforces global, local, and client fairness in multi-class FL. Experimental results show that our method outperforms existing baselines in terms of fairness–accuracy trade-off, communication efficiency and computational cost.

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

APPENDIX

## A  MOTIVATION EXAMPLE FOR MULTIPLE FAIRNESS CONCEPTS

We provide a real-world example with the corresponding law or regulation to demonstrate the case where multiple fairness concepts must be addressed simultaneously.

Consider a system used to assess eligibility for public assistance programs, such as food stamps, Medicaid, and federally funded education programs across U.S. states. Each state is a client in this setup. Title VI of the Civil Rights Act (1964) (United States, 1964) prohibits discrimination in federally funded programs based on race, color, or national origin. Title I funding under ESSA (United States Department of Education, 2015) is required to be allocated based on district-level need to reduce disparities between those districts. The two rules require that the system satisfy three fairness concepts simultaneously. Specifically:

Global fairness ensures that qualified individuals from protected groups (e.g., racial minorities) have equal chances of receiving benefits compared to unprotected groups, in alignment with federal civil rights protections.

Local fairness ensures that, within each state, qualified individuals from both protected and unprotected groups are treated equitably, in alignment with state-level civil rights requirements.

Client fairness ensures that individuals are not advantaged or disadvantaged based on their geographic location (e.g., district or state), aligning with the equity goals underlying Title I funding under ESSA.

## B  THE PARAMETERS OF CONVEX PROGRAM IN PROPOSITION 3.1

This section provides the parameters used in the convex program in Proposition 3.1.

**Objective:**  We formulate the objective as the negative accuracy, which we aim to minimize:

$$-\mathrm{Pr}_D(\widetilde{Y} = Y) = -\sum_{\mathcal{Y},\mathcal{A},\mathcal{C}} \mathrm{Pr}_D(Y = y, A = a, C = c) \cdot \mathrm{Pr}_D(\widetilde{Y} = y \mid Y = y, A = a, C = c) = -\sum_{\mathcal{Y},\mathcal{A},\mathcal{C}} p_{ac}^y \cdot z_{ac}^y,$$

where $p_{ac}^y = \mathrm{Pr}_D(Y = y, A = a, C = c)$ and $z_{ac}^y$ is the variable representing the true positives of $\widetilde{Y}$ for class $y$, group $a$ and client $c$.

This yields the objective vector:

$$\mathbf{c}^T = \begin{bmatrix} \mathbf{c}_{01}^T & \mathbf{c}_{11}^T & \mathbf{c}_{02}^T & \mathbf{c}_{12}^T & \cdots & \mathbf{c}_{0K}^T & \mathbf{c}_{1K}^T \end{bmatrix} \in \mathbb{R}^{2NK}$$

where each block is defined as:

$$\mathbf{c}_{ac}^T = \begin{bmatrix} -p_{ac}^1 & -p_{ac}^2 & \cdots & -p_{ac}^N \end{bmatrix} \in \mathbb{R}^N$$

**Fairness Constraints:**  The constraints are represented as:

$$-\mathbf{b} \le \mathbf{A}\mathbf{z} \le \mathbf{b},$$

with the matrix $\mathbf{A}$ and vector $\mathbf{b}$ defined as the vertical stacking of all constraint components.

**Global Group Fairness:** For each $y \in \mathcal{Y}$, global group fairness requires:

$$\left| \mathrm{Pr}_D(\widetilde{Y} = y \mid Y = y, A = 1) - \mathrm{Pr}_D(\widetilde{Y} = y \mid Y = y, A = 0) \right| \le \epsilon_g.$$

where, the term $\mathrm{Pr}_D(\widetilde{Y} = y | Y = y, A = a)$ is equal to:

$$\mathrm{Pr}_D(\widetilde{Y} = y | Y = y, A = a) = \sum_{\mathcal{C}} \mathrm{Pr}(\widetilde{Y} = y | Y = y, A = a, C = c) \cdot \mathrm{Pr}_D(C = c | Y = y, A = a)$$

$$= \sum_{\mathcal{C}} \mathrm{Pr}(\widetilde{Y} = y | Y = y, A = a, C = c) \cdot \frac{\mathrm{Pr}_D(C = c, Y = y, A = a)}{\mathrm{Pr}(Y = y, A = a)}$$

$$= \sum_{\mathcal{C}} \frac{z_{ac}^y \cdot p_{ac}^y}{\alpha_a^y}$$

$$\tag{12}$$

with,

$$p_{ac}^y = \Pr_D(Y = y, A = a, C = c)$$
$$\alpha_a^y = \Pr_D(Y = y, A = a)$$

Define:

$$\mathbf{v}_{ac}^y = \frac{p_{ac}^y}{\alpha_a^y} \cdot \mathbf{e}^y \in \mathbb{R}^N,$$

where $\mathbf{e}^y$ is the $y$-th canonical basis vector. Then define:

$$\mathbf{V}_c = \begin{bmatrix} (\mathbf{v}_{0c}^1)^T & -(\mathbf{v}_{1c}^1)^T \\ \vdots & \vdots \\ (\mathbf{v}_{0c}^N)^T & -(\mathbf{v}_{1c}^N)^T \end{bmatrix} \in \mathbb{R}^{N \times 2N},$$

The global group fairness matrix is:

$$\mathbf{A}_1 = [\mathbf{V}_1, \mathbf{V}_2, \ldots, \mathbf{V}_K] \in \mathbb{R}^{N \times 2NK}, \quad \mathbf{b}_1 = \epsilon_g \cdot \mathbf{1}_N.$$

**Local Group Fairness:** Local group fairness constraint is for all $c \in \mathcal{C}, y \in \mathcal{Y}$:

$$\left| \Pr_D(\widetilde{Y} = y | Y = y, C = c, A = 1) - \Pr_D(\widetilde{Y} = y | Y = y, C = c, A = 0) \right| \leq \epsilon_l$$

$$\rightleftharpoons |z_{0c}^y - z_{1c}^y| \leq \epsilon_l.$$

Define:

$$\mathbf{M} = \begin{bmatrix} (\mathbf{e}^1)^T & -(\mathbf{e}^1)^T \\ \vdots & \vdots \\ (\mathbf{e}^N)^T & -(\mathbf{e}^N)^T \end{bmatrix} \in \mathbb{R}^{N \times 2N},$$

then, the matrix for local group fairness is:

$$\mathbf{A}_2 = \begin{bmatrix} \mathbf{M} & \mathbf{0} & \cdots & \mathbf{0} \\ \mathbf{0} & \mathbf{M} & \cdots & \mathbf{0} \\ \vdots & \vdots & \ddots & \vdots \\ \mathbf{0} & \mathbf{0} & \cdots & \mathbf{M} \end{bmatrix} \in \mathbb{R}^{NK \times 2NK}, \quad \mathbf{b}_2 = \epsilon_l \cdot \mathbf{1}_{NK}.$$

**Client Fairness:** Client fairness constraint is for all $c \in \mathcal{C}$:

$$\left| \Pr_D(\widetilde{Y} = Y \mid C = c) - \Pr_D(\widetilde{Y} = Y) \right| \leq \epsilon_c,$$

where:

$$\Pr_D(\widetilde{Y} = Y \mid C = c) = \sum_{\mathcal{Y}, \mathcal{A}} \frac{p_{ac}^y}{p_c} \cdot z_{ac}^y, \quad \Pr_D(\widetilde{Y} = Y) = \sum_{\mathcal{Y}, \mathcal{A}, \mathcal{C}} p_{ac}^y \cdot z_{ac}^y,$$

and $p_c = \Pr_D(C = c)$.

Define:

$$\mathbf{w}_{ac} = \left[ \frac{-p_{ac}^1}{p_c}, \frac{-p_{ac}^2}{p_c}, \ldots, \frac{-p_{ac}^N}{p_c} \right]^T \in \mathbb{R}^N,$$

and construct $\mathbf{A}_3 \in \mathbb{R}^{K \times 2NK}$ as:

$$\mathbf{A}_3 = \begin{bmatrix} -(\mathbf{c}_{01} + \mathbf{w}_{01}) & -(\mathbf{c}_{11} + \mathbf{w}_{11}) & \cdots & \mathbf{c}_{0K} & \mathbf{c}_{1K} \\ \mathbf{c}_{01} & \mathbf{c}_{11} & -(\mathbf{c}_{02} + \mathbf{w}_{02}) & \cdots & \mathbf{c}_{1K} \\ \vdots & \vdots & \vdots & \ddots & \vdots \\ \mathbf{c}_{01} & \mathbf{c}_{11} & \cdots & -(\mathbf{c}_{0K} + \mathbf{w}_{0K}) & -(\mathbf{c}_{1K} + \mathbf{w}_{1K}) \end{bmatrix}, \quad \mathbf{b}_3 = \epsilon_c \cdot \mathbf{1}_K.$$

**Full Convex Program:** The full convex program is: The convex program equation 6 is

$$
\begin{aligned}
&\text{minimize:} && \mathbf{c}^T \mathbf{z} \\
&\text{with respect to:} && \mathbf{z} \in \mathbb{R}^{2NK} \\
&\text{subject to:} && -\mathbf{b} \leq \mathbf{A}\mathbf{z} \leq \mathbf{b} \\
& && \mathbf{z}_{ac} \in D_{ac}, \forall a \in \mathcal{A}, c \in \mathcal{C}
\end{aligned}
\tag{13}
$$

with:

$$
\mathbf{A} = \begin{bmatrix} \mathbf{A}_1 \\ \mathbf{A}_2 \\ \mathbf{A}_3 \end{bmatrix} \in \mathbb{R}^{(N+NK+K)\times 2NK}, \quad \mathbf{b} = \begin{bmatrix} \mathbf{b}_1 \\ \mathbf{b}_2 \\ \mathbf{b}_3 \end{bmatrix} \in \mathbb{R}^{N+NK+K}
$$

$D_{ac}$ is the convex set defined in Def. 2.5.

## C  EXTENSION TO OTHER FAIRNESS METRICS (EQUAL OPPORTUNITY, EQUALIZED ODDS, DISPARATE MISTREATMENT AND STATISTICAL PARITY)

This section provides details on how to modify the inequality constraints of the convex program equation 6 so that the framework can be extended to other fairness metrics. For each metric, we use a distributive fairness concept to demonstrate that the corresponding fairness constraint is a linear constraint with respect to the variables $z_{ac}^y$. The parameters of those inequality constraints are statistics derived from the FL system $D$.

### C.1  EQUAL OPPORTUNITY

(1). $\epsilon_g$- global group fairness:

$$
\left| \mathrm{Pr}_D(\widetilde{Y} = 1 | Y = 1, A = 1) - \mathrm{Pr}_D(\widetilde{Y} = 1 | Y = 1, A = 0) \right| \leq \epsilon_g
$$

(2). $\epsilon_l$- local group fairness:

$$
\left| \mathrm{Pr}_D(\widetilde{Y} = 1 | Y = 1, A = 1, C = c) - \mathrm{Pr}_D(\widetilde{Y} = 1 | Y = 1, A = 0, C = c) \right| \leq \epsilon_l, \forall c \in \mathcal{C}
$$

(3). $\epsilon_c$- client fairness:

$$
\left| \mathrm{Pr}_D(\widetilde{Y} = 1 | Y = 1, C = c) - \mathrm{Pr}_D(\widetilde{Y} = 1 | Y = 1) \right| \leq \epsilon_c, \forall c \in \mathcal{C}
$$

We use global group fairness to illustrate that the equal opportunity constraint can be expressed as a linear inequality with respect to the variables. The constraints for local and client fairness follow in a similar manner.

The global group fairness constraints is:

$$
|\mathrm{Pr}_D(\widetilde{Y} = 1 | Y = 1, A = 1) - \mathrm{Pr}_D(\widetilde{Y} = 1 | Y = 1, A = 0)| \leq \epsilon_g
$$

$$
\rightleftharpoons -\epsilon_g \leq \sum_{\mathcal{C}} \mathrm{Pr}_D(\widetilde{Y} = 1 | Y = 1, A = 1, C = c) \cdot \mathrm{Pr}_D(C = c | Y = 1, A = 1)
$$

$$
- \sum_{\mathcal{C}} \mathrm{Pr}_D(\widetilde{Y} = 1 | Y = 1, A = 0, C = c) \cdot \mathrm{Pr}_D(C = c | Y = 1, A = 0) \leq \epsilon_g
$$

$$
\rightleftharpoons -\epsilon_g \leq \sum_{\mathcal{C}} z_{1c}^1 \cdot \mathrm{Pr}_D(C = c | Y = 1, A = 1) - \sum_{\mathcal{C}} z_{0c}^1 \cdot \mathrm{Pr}_D(C = c | Y = 1, A = 0) \leq \epsilon_g
$$

### C.2  EQUALIZED ODDS

This section discusses fairness under the multi-class Equal Opportunity, which generalizes Equalized Odds for the binary-class settings in Hardt et al. (2016).

(1) $\epsilon_g$- global group fairness:

$$\left| \Pr_D(\widetilde{Y} = y | Y = y, A = 1) - \Pr_D(\widetilde{Y} = y | Y = y, A = 0) \right| \le \epsilon_g, \forall y \in \mathcal{Y}$$

(2) $\epsilon_l$- local group fairness:

$$\left| \Pr_D(\widetilde{Y} = y | Y = y, A = 1, C = c) - \Pr_D(\widetilde{Y} = y | Y = y, A = 0, C = c) \right| \le \epsilon_l, \forall y \in \mathcal{Y}, c \in \mathcal{C}$$

(3) $\epsilon_c$- client fairness:

$$\left| \Pr_D(\widetilde{Y} = y | Y = y, C = c) - \Pr_D(\widetilde{Y} = y | Y = y) \right| \le \epsilon_c, \forall y \in \mathcal{Y}, c \in \mathcal{C}$$

We use client fairness to illustrate that the Equal Opportunity constraint can be formulated as a linear inequality w.r.t our variables. The cases of global and local group fairness follow in a similar manner.

The client fairness constraint is:

$$|\Pr_D(\widetilde{Y} = y | Y = y, C = c) - \Pr_D(\widetilde{Y} = y | Y = y)| \le \epsilon_c$$

$$\rightleftharpoons -\epsilon_c \le \sum_{\mathcal{A}} \Pr_D(\widetilde{Y} = y | Y = y, C = c, A = a) \cdot \Pr_D(A = a | Y = y, C = c)$$

$$- \sum_{\mathcal{A},\mathcal{C}} \Pr_D(\widetilde{Y} = y | Y = y, C = c, A = a) \cdot \Pr_D(A = a, C = c | Y = y) \le \epsilon_c$$

$$\rightleftharpoons -\epsilon_c \le \sum_{\mathcal{A}} z_{ac}^y \cdot \Pr_D(A = a | Y = y, C = c) - \sum_{\mathcal{A},\mathcal{C}} z_{ac}^y \cdot \Pr_D(A = a, C = c | Y = y) \le \epsilon_c$$

## C.3 DISPARATE MISTREATMENT

(1) $\epsilon_g$- global group fairness:

$$\left| \Pr_D(\widetilde{Y} = Y | A = 1) - \Pr_D(\widetilde{Y} = Y | A = 0) \right| \le \epsilon_g$$

(2) $\epsilon_l$- local group fairness:

$$\left| \Pr_D(\widetilde{Y} = Y | A = 1, C = c) - \Pr_D(\widetilde{Y} = Y | A = 0, C = c) \right| \le \epsilon_l, \forall c \in \mathcal{C}$$

(3) $\epsilon_c$- client fairness:

$$\left| \Pr_D(\widetilde{Y} = Y | C = c) - \Pr_D(\widetilde{Y} = Y) \right| \le \epsilon_c, \forall c \in \mathcal{C}$$

We use local group fairness, the other fairness concepts follows similarly.

$$|\Pr_D(\widetilde{Y} = Y | A = 1, C = c) - \Pr_D(\widetilde{Y} = Y | A = 0, C = c)| \le \epsilon_l$$

$$\rightleftharpoons -\epsilon_l \le \sum_{\mathcal{Y}} \Pr_D(\widetilde{Y} = y | Y = y, A = 1, C = c) \cdot \Pr_D(Y = y | A = 1, C = c)$$

$$- \sum_{\mathcal{Y}} \Pr_D(\widetilde{Y} = y | Y = y, A = 0, C = c) \cdot \Pr_D(Y = y | A = 0, C = c) \le \epsilon_l$$

$$\rightleftharpoons -\epsilon_l \le \sum_{\mathcal{Y}} \le z_{1c}^y \Pr_D(Y = y | A = 1, C = c) - \sum_{\mathcal{Y}} \le z_{1c}^y \Pr_D(Y = y | A = 1, C = c) \le \epsilon_l$$

$$\tag{14}$$

## C.4 STATISTICAL PARITY

(1).$\epsilon_g$-global group fairness:

$$\left| \Pr_D(\widetilde{Y} = y | A = 1) - \Pr_D(\widetilde{Y} = y | A = 0) \right| \le \epsilon_g, \forall y \in \mathcal{Y}$$

(2). $\epsilon_l$-local group fairness:

$$\left| \Pr_D(\widetilde{Y} = y | A = 1, C = c) - \Pr_D(\widetilde{Y} = y | A = 0, C = c) \right| \le \epsilon_l, \quad \forall y \in \mathcal{Y}, c \in \mathcal{C}$$

(3).$\epsilon_c$-client fairness:

$$\left| \Pr_D(\widetilde{Y} = y | C = c) - \Pr_D(\widetilde{Y} = y) \right| \le \epsilon_c, \quad \forall y \in \mathcal{Y}, c \in \mathcal{C}$$

The variables we use to characterize the LP for multi-class statistical parity differ from those used for Equalized Odds and Equal Opportunity. Statistical Parity requires that the predictor's positive rate be the same for both sensitive and non-sensitive groups. Therefore, we need to focus on both true positives and false positives for a given class.

In the statistical parity setting, let $\widetilde{Y}_{\boldsymbol{\theta_1}}$ be the *derived outcome predictor* derived by the uniform vector $\boldsymbol{\theta_1}$, the variables we use to characterize the outcome predictor $\widetilde{Y} : \mathcal{X} \times \mathcal{A} \times \mathcal{C} \to \mathcal{Y}$ are:

$$z_{ac}^{yj} = \Pr(\widetilde{Y} = y | \widetilde{Y}_{\boldsymbol{\theta_1}} = j, A = a, C = c) \tag{15}$$

We define the following statistics:

$$u_{ac}^{yj} = \Pr_D(Y = y, \widetilde{Y}_{\boldsymbol{\theta_1}} = j, A = a, C = c)$$
$$u_{ac}^{j} = \Pr_D(\widetilde{Y}_{\boldsymbol{\theta_1}} = j, A = a, C = c)$$
$$u_a = \Pr_D(A = a)$$

The objective function we maximize is:

$$\Pr_D(\widetilde{Y} = Y) = \sum_{y \in \mathcal{Y}} \Pr(\widetilde{Y} = y, Y = y)$$

$$= \sum_{c \in \mathcal{C}} \sum_{a \in \mathcal{A}} \sum_{j \in \mathcal{Y}} \sum_{y \in \mathcal{Y}} \Pr(\widetilde{Y} = y, \widetilde{Y}_{\boldsymbol{\theta_1}} = j, A = a, C = c, Y = y)$$

$$= \sum_{c \in \mathcal{C}} \sum_{a \in \mathcal{A}} \sum_{j \in \mathcal{Y}} \sum_{y \in \mathcal{Y}} \Pr(\widetilde{Y} = y | \widetilde{Y}_{\boldsymbol{\theta_1}} = j, A = a, C = c, Y = y) \cdot \Pr_D(Y = y, \widetilde{Y}_{\boldsymbol{\theta_1}} = j, A = a, C = c)$$

$$= \sum_{c \in \mathcal{C}} \sum_{a \in \mathcal{A}} \sum_{j \in \mathcal{Y}} \sum_{y \in \mathcal{Y}} \Pr(\widetilde{Y} = y | \widetilde{Y}_{\boldsymbol{\theta_1}} = j, A = a, C = c) \cdot \Pr_D(Y = y, \widetilde{Y}_{\boldsymbol{\theta_1}} = j, A = a, C = c)$$

$$= \sum_{c \in \mathcal{C}} \sum_{a \in \mathcal{A}} \sum_{j \in \mathcal{Y}} \sum_{y \in \mathcal{Y}} u_{ac}^{yj} z_{ac}^{yj}$$

$$\tag{16}$$

The global group fairness is a linear inequality with respect to the defined variables

$$-\epsilon_g \le \Pr_D(\widetilde{Y} = y | A = 0) - \Pr_D(\widetilde{Y} = y | A = 1) \le \epsilon_g$$

$$\iff -\epsilon_g \le \frac{\Pr_D(\widetilde{Y} = y, A = 0)}{\Pr_D(A = 0)} - \frac{\Pr_D(\widetilde{Y} = y, A = 1)}{\Pr_D(A = 1)} \le \epsilon_g$$

$$\iff -\epsilon_g \le \sum_{c \in \mathcal{C}} \sum_{j \in \mathcal{Y}} \frac{\Pr_D(\widetilde{Y} = y | \widetilde{Y}_{\boldsymbol{\theta_1}} = j, C = c, A = 0) \cdot \Pr_D(\widetilde{Y}_{\boldsymbol{\theta_1}} = j, A = 0, C = c)}{\Pr_D(A = 0)}$$

$$- \sum_{c \in \mathcal{C}} \sum_{j \in \mathcal{Y}} \frac{\Pr_D(\widetilde{Y} = y | \widetilde{Y}_{\boldsymbol{\theta_1}} = j, C = c, A = 1) \cdot \Pr_D(\widetilde{Y}_{\boldsymbol{\theta_1}} = j, A = 1, C = c)}{\Pr_D(A = 1)} \le \epsilon_g$$

$$\iff -\epsilon_g \le \sum_{c \in \mathcal{C}} \sum_{j \in \mathcal{Y}} \frac{z_{0c}^{yj} \cdot u_{0c}^{j}}{u_0} - \sum_{c \in \mathcal{C}} \sum_{j \in \mathcal{Y}} \frac{z_{1c}^{yj} \cdot u_{1c}^{j}}{u_1} \le \epsilon_g$$

$$\tag{17}$$

# D THE PARAMETERS OF LP EQUATION 8

The parameters of the LP are identical to those of the convex program equation 6, except that the condition requiring $\mathbf{z}_{ac}$ to lie in a convex set $D_{ac}$ is approximated by a set of linear inequalities, $\mathbf{K}_{ac}\mathbf{z}_{ac} \leq \mathbf{l}_{ac}$, which are detailed below.

# E THE PARAMETERS OF SIMPLEX $\widehat{D_{ac}}$

The N-dimensional polytope $\widehat{D_{ac}}$ can be defined using (N+1) inequalities. Any single point $\mathbf{u} \in \mathbb{R}^N$ lies in the $\widehat{D_{ac}}$ must have:

$$\mathbf{K}_{ac}\mathbf{u} \leq \mathbf{l}_{ac} \tag{18}$$

then,

$$\mathbf{K}_{ac} = \begin{bmatrix} -1 & -1 & -1 & \cdots & -1 \\ 1 - \sum_{i \in \mathcal{Y}, i \neq 1} \mathrm{TP}_{ac}^i(\widetilde{Y}_{\boldsymbol{\theta_1}}) & \mathrm{TP}_{ac}^1(\widetilde{Y}_{\boldsymbol{\theta_1}}) & \mathrm{TP}_{ac}^1(\widetilde{Y}_{\boldsymbol{\theta_1}}) & \cdots & \mathrm{TP}_{ac}^1(\widetilde{Y}_{\boldsymbol{\theta_1}}) \\ \mathrm{TP}_{ac}^2(\widetilde{Y}_{\boldsymbol{\theta_1}}) & 1 - \sum_{i \in \mathcal{Y}, i \neq 2} \mathrm{TP}_{ac}^i(\widetilde{Y}_{\boldsymbol{\theta_1}}) & \mathrm{TP}_{ac}^2(\widetilde{Y}_{\boldsymbol{\theta_1}}) & \cdots & \mathrm{TP}_{ac}^2(\widetilde{Y}_{\boldsymbol{\theta_1}}) \\ \mathrm{TP}_{ac}^3(\widetilde{Y}_{\boldsymbol{\theta_1}}) & \mathrm{TP}_{ac}^3(\widetilde{Y}_{\boldsymbol{\theta_1}}) & 1 - \sum_{i \in \mathcal{Y}, i \neq 3} \mathrm{TP}_{ac}^i(\widetilde{Y}_{\boldsymbol{\theta_1}}) & \cdots & \mathrm{TP}_{ac}^3(\widetilde{Y}_{\boldsymbol{\theta_1}}) \\ \vdots & \vdots & \vdots & \ddots & \vdots \\ \mathrm{TP}_{ac}^N(\widetilde{Y}_{\boldsymbol{\theta_1}}) & \mathrm{TP}_{ac}^N(\widetilde{Y}_{\boldsymbol{\theta_1}}) & \mathrm{TP}_{ac}^N(\widetilde{Y}_{\boldsymbol{\theta_1}}) & \cdots & 1 - \sum_{i \in \mathcal{Y}, i \neq N} \mathrm{TP}_{ac}^i(\widetilde{Y}_{\boldsymbol{\theta_1}}) \end{bmatrix} \in \mathbb{R}^{(N+1) \times N}$$

$$\mathbf{l}_{ac} = \begin{bmatrix} -1 & \mathrm{TP}_{ac}^1(\widetilde{Y}_{\boldsymbol{\theta_1}}) & \mathrm{TP}_{ac}^2(\widetilde{Y}_{\boldsymbol{\theta_1}}) & \mathrm{TP}_{ac}^3(\widetilde{Y}_{\boldsymbol{\theta_1}}) & \cdots & \mathrm{TP}_{ac}^N(\widetilde{Y}_{\boldsymbol{\theta_1}}) \end{bmatrix}^T \in \mathbb{R}^{N+1}$$

# F THE PARAMETERS OF LAE IN PROPOSITION 3.2

The parameters of LAE equation 9: $\mathbf{G}_{ac}\boldsymbol{\beta}_{ac} = \boldsymbol{\gamma}_{ac}$ are:

$$\mathbf{G}_{ac} = \begin{bmatrix} 1 & 1 & 1 & \cdots & 1 & 1 \\ \mathrm{TP}_{ac}^1(\widetilde{Y}_{\boldsymbol{\theta_1}}) & 1 & 0 & \cdots & 0 & 0 \\ \mathrm{TP}_{ac}^2(\widetilde{Y}_{\boldsymbol{\theta_1}}) & 0 & 1 & \cdots & 0 & 0 \\ \vdots & \vdots & \vdots & \ddots & \vdots & \vdots \\ \mathrm{TP}_{ac}^{N-1}(\widetilde{Y}_{\boldsymbol{\theta_1}}) & 0 & 0 & \cdots & 1 & 0 \\ \mathrm{TP}_{ac}^N(\widetilde{Y}_{\boldsymbol{\theta_1}}) & 0 & 0 & \cdots & 0 & 1 \end{bmatrix}, \quad \boldsymbol{\gamma}_{ac} = \begin{bmatrix} 1 \\ \mathbf{z}_{ac} \end{bmatrix} \tag{19}$$

# G THEORETICAL PROOFS

## G.1 PROOF OF PROPOSITION 2.6

**Proposition 2.6:** Let $D_{ac}$ be the region defined in Def.2.5. Then, $D_{ac}$ is a convex set. For any predictor $\widetilde{Y} : \mathcal{X} \times \mathcal{A} \times \mathcal{C} \to \mathcal{Y}$, let the point representing true positives of $\widetilde{Y}$ be: $\mathbf{TP}_{ac}(\widetilde{Y}) = [\mathrm{TP}_{ac}^1(\widetilde{Y}), \cdots, \mathrm{TP}_{ac}^N(\widetilde{Y})]$. Then, $\mathbf{TP}_{ac}(\widetilde{Y})$ lies in $D_{ac}$.

**Proof:** We first show that $D_{ac}$ is convex. Consider any $\mathbf{r}_0, \mathbf{r}_1 \in [0,1]^N$, from Def. 2.5, if $\mathbf{r}_0, \mathbf{r}_1 \in D_{ac}$, then:

$$\begin{aligned} \mathbf{v}_{\boldsymbol{\theta}}^T \mathbf{r}_0 &\leq \mathbf{v}_{\boldsymbol{\theta}}^T \mathbf{TP}_{ac}(\widetilde{Y}_{\boldsymbol{\theta}}), \quad \forall \boldsymbol{\theta} \in \mathbb{R}_{\geq 0}^N, \\ \mathbf{v}_{\boldsymbol{\theta}}^T \mathbf{r}_1 &\leq \mathbf{v}_{\boldsymbol{\theta}}^T \mathbf{TP}_{ac}(\widetilde{Y}_{\boldsymbol{\theta}}), \quad \forall \boldsymbol{\theta} \in \mathbb{R}_{\geq 0}^N. \end{aligned} \tag{20}$$

Since $\lambda \in [0, 1]$, we must have:

$$
\begin{aligned}
\mathbf{v}_{\boldsymbol{\theta}}^T(\lambda \mathbf{r}_0) &\leq \mathbf{v}_{\boldsymbol{\theta}}^T(\lambda \mathbf{TP}_{ac}(\widetilde{Y}_{\boldsymbol{\theta}})), \quad \forall \boldsymbol{\theta} \in \mathbb{R}_{\geq 0}^N, \\
\mathbf{v}_{\boldsymbol{\theta}}^T((1-\lambda)\mathbf{r}_1) &\leq \mathbf{v}_{\boldsymbol{\theta}}^T((1-\lambda)\mathbf{TP}_{ac}(\widetilde{Y}_{\boldsymbol{\theta}})), \quad \forall \boldsymbol{\theta} \in \mathbb{R}_{\geq 0}^N, \\
\Rightarrow \mathbf{v}_{\boldsymbol{\theta}}^T(\lambda \mathbf{r}_0 + (1-\lambda)\mathbf{r}_1) &\leq \mathbf{v}_{\boldsymbol{\theta}}^T \mathbf{TP}_{ac}(\widetilde{Y}_{\boldsymbol{\theta}}), \quad \forall \boldsymbol{\theta} \in \mathbb{R}_{\geq 0}^N,
\end{aligned}
\tag{21}
$$

which implies: $\lambda \mathbf{r}_0 + (1-\lambda)\mathbf{r}_1 \in D_{ac}$.

We choose $\mathbf{r}_0, \mathbf{r}_1 \in D_{ac}$ arbitrarily. Thus, for all $\mathbf{r}_0, \mathbf{r}_1 \in D_{ac}$ and $\lambda \in [0, 1]$, it holds that $\lambda \mathbf{r}_0 + (1-\lambda)\mathbf{r}_1 \in D_{ac}$. Therefore, $D_{ac}$ is a convex set.

We then show that for any predictor $\widetilde{Y} : \mathcal{X} \times \mathcal{A} \times \mathcal{C} \to \mathcal{Y}$, let the point representing true positives of $\widetilde{Y}$ be: $\mathbf{TP}_{ac}(\widetilde{Y}) = [\mathrm{TP}_{ac}^y(\widetilde{Y}), \cdots, \mathrm{TP}_{ac}^N(\widetilde{Y})]^T$. Then, $\mathbf{TP}_{ac}(\widetilde{Y})$ lies in $D_{ac}$.

$$
\begin{aligned}
\mathbf{TP}_{ac}(\widetilde{Y}) \in D_{ac} &\iff \mathbf{TP}_{ac}(\widetilde{Y}) \in \bigcap_{\boldsymbol{\theta} \in \mathbb{R}_{\geq 0}^N} \left\{ \mathbf{x} \in [0, 1]^N | \mathbf{v}_{\boldsymbol{\theta}}^T \mathbf{x} \leq \mathbf{v}_{\boldsymbol{\theta}}^T \mathbf{TP}_{ac}(\widetilde{Y}_{\boldsymbol{\theta}}) \right\} \\
&\iff \forall \boldsymbol{\theta} \in \mathbb{R}_{\geq 0}^N, \quad \mathbf{v}_{\boldsymbol{\theta}}^T \mathbf{TP}_{ac}(\widetilde{Y}) \leq \mathbf{v}_{\boldsymbol{\theta}}^T \mathbf{TP}_{ac}(\widetilde{Y}_{\boldsymbol{\theta}})
\end{aligned}
\tag{22}
$$

To prove this, we consider the value of $\mathbf{v}_{\boldsymbol{\theta}}^T \mathbf{TP}_{ac}(\widetilde{Y})$:

$$
\begin{aligned}
\mathbf{v}_{\boldsymbol{\theta}}^T \mathbf{TP}_{ac}(\widetilde{Y}) &= \sum_{y \in \mathcal{Y}} \theta_y \mathrm{Pr}_D(Y = y \mid A = a, C = c) \mathrm{TP}_{ac}^y(\widetilde{Y}) \\
&= \sum_{y \in \mathcal{Y}} \theta_y \mathrm{Pr}_D(Y = y \mid A = a, C = c) \mathbb{E}_{\mathrm{Pr}_{X|A,C}}[\mathbf{1}(\widetilde{Y} = y) \cdot r_y(X, a, c)] \cdot \frac{1}{\mathrm{Pr}_D(Y = y \mid A = a, C = c)} \\
&= \sum_{y \in \mathcal{Y}} \theta_y \mathbb{E}_{\mathrm{Pr}_{X|A,C}}[\mathbf{1}(\widetilde{Y} = y) \cdot r_y(X, a, c)] \\
&= \sum_{y \in \mathcal{Y}} \mathbb{E}_{\mathrm{Pr}_{X|A,C}}[\mathbf{1}(\widetilde{Y} = y) \cdot \theta_y r_y(X, a, c)]
\end{aligned}
\tag{23}
$$

Equation equation 23 achieves the maximum value if:

$$
\widetilde{Y} = y, \quad \text{if } \theta_y r_y(x, a, c) = \max_{i=1}^N \theta_i r_i(x, a, c)
\tag{24}
$$

The predictor that takes the value of Eq. 24 is a derived outcome predictor $\widetilde{Y}_{\boldsymbol{\theta}}$ as defined in Def. 2.3. The derived outcome predictor $\widetilde{Y}_{\boldsymbol{\theta}}$ maximizes the value of $\mathbf{v}_{\boldsymbol{\theta}}^T \mathbf{TP}_{ac}(\widetilde{Y})$ for all $\boldsymbol{\theta} \in \mathbb{R}_{\geq 0}^N$. Therefore, any predictor must satisfy $\mathbf{v}_{\boldsymbol{\theta}}^T \mathbf{TP}_{ac}(\widetilde{Y}) \leq \mathbf{v}_{\boldsymbol{\theta}}^T \mathbf{TP}_{ac}(\widetilde{Y}_{\boldsymbol{\theta}})$, which is equivalent to $\mathbf{TP}_{ac}(\widetilde{Y}) \in D_{ac}$.

## G.2 PROOF OF PROPOSITION 3.1

**Proposition 3.1:** Let the vector $\mathbf{z} \in \mathbb{R}^{2NK}$:

$$
\begin{aligned}
\mathbf{z}^T &= \begin{bmatrix} \mathbf{z}_{01}^T & \mathbf{z}_{11}^T & \mathbf{z}_{02}^T & \mathbf{z}_{12}^T \cdots & \mathbf{z}_{0K}^T & \mathbf{z}_{1K}^T \end{bmatrix}, \\
\text{with,} \quad \mathbf{z}_{ac}^T &= \begin{bmatrix} z_{ac}^1 & z_{ac}^2 & z_{ac}^3 & \cdots & z_{ac}^N \end{bmatrix} \in \mathbb{R}^N
\end{aligned}
$$

satisfy the following convex program

$$
\begin{aligned}
\text{minimize:} \quad & \mathbf{c}^T \mathbf{z} \\
\text{with respect to:} \quad & \mathbf{z} \in \mathbb{R}^{2NK} \\
\text{subject to:} \quad & -\mathbf{b} \leq \mathbf{A}\mathbf{z} \leq \mathbf{b} \\
& \mathbf{z}_{ac} \in D_{ac}, \forall a \in \mathcal{A}, c \in \mathcal{C}
\end{aligned}
\tag{25}
$$

then, the outcome predictor $\widetilde{Y} : \mathcal{X} \times \mathcal{A} \times \mathcal{C} \rightarrow \mathcal{Y}$ that satisfies eq. equation 7 for all $y \in \mathcal{Y}, a \in \mathcal{A}, c \in \mathcal{C}$

$$\Pr(\widetilde{Y} = y | Y = y, A = a, C = c) = z_{ac}^y \tag{26}$$

is a $\epsilon$-fair optimal outcome predictor. The optimal accuracy for a $\epsilon$-fair outcome predictor is $-\mathbf{c}^T \mathbf{z}$.

**Proof:** As discussed in Appendix B, the outcome predictor $\widetilde{Y}$ whose true positives satisfy the first $N$ constraints satisfies the $\epsilon_g$- global group fairness condition. The next $NK$ constraints represent the $\epsilon_l$- local group fairness, and the last $K$ constraints represent the client fairness constraints. Therefore, a classifier that satisfies Eq. equation 26 will satisfy all three distributive fairness concepts.

The objective function of the convex program is:

$$\mathbf{c}^T \mathbf{z}$$

$$= -\sum_{c \in \mathcal{C}} \sum_{a \in \mathcal{A}} \sum_{y \in \mathcal{Y}} z_{ac}^y p_{ac}^y$$

$$= -\sum_{c \in \mathcal{C}} \sum_{a \in \mathcal{A}} \sum_{y \in \mathcal{Y}} \Pr_D(\widetilde{Y} = y | Y = y, A = a, C = c) \Pr_D(Y = y, A = a, C = c)$$

$$= -\sum_{y \in \mathcal{Y}} \Pr_D(\widetilde{Y} = y, Y = y)$$

$$= -\Pr_D(\widetilde{Y} = Y)$$

The predictor $\widetilde{Y}$ that minimizes $\mathbf{c}^T \mathbf{z}$ corresponds to maximum accuracy. The maximum accuracy is $-\mathbf{c}^T \mathbf{z}$. The predictor $\widetilde{Y}$ that satisfies the convex program is an optimal $\epsilon$-outcome predictor. The accuracy of the optimal $\epsilon$-outcome predictor is $-\mathbf{c}^T \mathbf{z}$.

### G.3 PROOF OF PROPOSITION 3.2

**Proposition 3.2:** Let $\mathbf{z} \in \mathbb{R}^{2NK}$ be the solution of the LP (8)

$$\mathbf{z}^T = \begin{bmatrix} \mathbf{z}_{01}^T & \mathbf{z}_{11}^T & \mathbf{z}_{02}^T & \mathbf{z}_{12}^T \cdots & \mathbf{z}_{0K}^T & \mathbf{z}_{1K}^T \end{bmatrix}$$

and $\mathrm{TP}_{ac}^y(\widetilde{Y}_{\boldsymbol{\theta_1}})$ be the true positive of the *derived outcome predictor* by $\boldsymbol{\theta_1}$ For all $a \in \mathcal{A}, c \in \mathcal{C}$, let $\boldsymbol{\beta}_{ac} = [\beta_{ac}^0, \beta_{ac}^1, \cdots, \beta_{ac}^N]$ be the solution of the following linear algebraic equation (LAE),

$$\mathbf{G}_{ac} \boldsymbol{\beta}_{ac} = \boldsymbol{\gamma}_{ac} \tag{27}$$

where, the parameter $\mathbf{G}_{ac} \in \mathbb{R}^{(N+1) \times (N+1)}, \boldsymbol{\gamma}_{ac} \in \mathbb{R}^{N+1}$, are detailed in Appendix F. Then. the predictor $\widetilde{Y}_{\boldsymbol{\beta}_{ac}}$ that takes value,

$$\widetilde{Y}_{\boldsymbol{\beta}_{ac}}(x, a, c) = \begin{cases} \widetilde{Y}_{\boldsymbol{\theta_1}}(x, a, c), & \text{with the probability } \beta_{ac}^0 \\ y, & \text{with the probability } \beta_{ac}^y, \forall y \in \mathcal{Y} \end{cases} \tag{28}$$

is a fair outcome predictor. There always exists a unique set of parameters $\{\boldsymbol{\beta}_{ac}\}_{\mathcal{A},\mathcal{C}}$, where $\boldsymbol{\beta}_{ac} \in [0, 1]^{N+1}$ and $|\boldsymbol{\beta}_{ac}|_{\ell_1} = 1$ that satisfies the LAE.

**Proof:** The true positive of class $y$ for the outcome predictor $\widetilde{Y}_{\boldsymbol{\beta}_{ac}} : \mathcal{X} \times \mathcal{A} \times \mathcal{C} \rightarrow \mathcal{Y}$ that takes value of Eq. equation 28 is:

$$\mathrm{TP}_{ac}^y(\widetilde{Y}_{\boldsymbol{\beta}_{ac}}) = \mathrm{TP}_{ac}^y(\widetilde{Y}_{\boldsymbol{\theta_1}})\beta_{ac}^0 + \beta_{ac}^y \tag{29}$$

Since $\boldsymbol{\beta}_{ac}$ is the solution of LAE equation 9, its elements satisfies that: $\forall y \in \mathcal{Y}$,

$$\mathrm{TP}_{ac}^y(\widetilde{Y}_{\boldsymbol{\theta_1}})\beta_{ac}^0 + \beta_{ac}^y = z_{ac}^y$$
$$\iff \mathrm{TP}_{ac}^y(\widetilde{Y}_{\boldsymbol{\beta}_{ac}}) = z_{ac}^y \tag{30}$$

The true positives of the predictor $\widetilde{Y}_{\boldsymbol{\beta}ac}$ for class $y$, client $c$, and group $a$ are $z_{ac}^y$. Since $\{z_{ac}^y\}_{\mathcal{A},\mathcal{C},\mathcal{Y}}$ is the solution of LP equation 8, it satisfies the fairness constraints of LP equation 8. Thus, the predictor $\widetilde{Y}_{\boldsymbol{\beta}ac}$ is a fair outcome predictor.

$\mathbf{G}_{ac}$ is a full-rank matrix for all clients and groups, thus, the LAE has a unique solution in all clients and groups.

### G.4 THE SOLUTION OF THE CONVEX PROGRAM EQUATION 6 ALWAYS EXISTS

A convex optimization problem has a solution if the feasible set is non-empty, compact and the objective function is continuous.

First, we show that the feasible set is non-empty. Consider a naive predictor $\widetilde{Y} : \mathcal{X} \times \mathcal{A} \times \mathcal{C} \to \mathcal{Y}$ whose true positives is a constant $r$ for all classes $y \in \mathcal{Y}$, clients $c \in \mathcal{C}$, and sensitive attributes $a \in \mathcal{A}$, i.e.,

$$\Pr_D(\widetilde{Y} = y \mid Y = y, A = a, C = c) = r, \forall y \in \mathcal{Y}, a \in \mathcal{A}, c \in \mathcal{C}$$

It is easy to verify that this naive classifier satisfies all distributive fairness concepts and lies within the convex set $D_{ac}$. Therefore, the feasible set of the convex program is non-empty.

Next, the feasible region is convex (Proposition 2.6), so it is compact. The objective function $\mathbf{c}^T \mathbf{z}$ is a linear function and therefore continuous. Thus, the convex program admits a solution.

There are always exists a predictor that can satisfies local, global and client fairness, for example, a predictor that outputs $y$ with probability $r$ satisfies all fairness concepts under Statistical Parity. A predictor with constant true positives,

$$\Pr_D(\tilde{Y} = y \mid Y = y, A = a, C = c) = c, \quad \forall y \in \mathcal{Y}, \ a \in \mathcal{A}, \ c \in \mathcal{C}$$

satisfies all fairness under Equal Opportunity. For any FL setup, there is at least one predictor that satisfies all fairness concepts. Our framework gives the one that satisfies fairness with optimal accuracy.

## H RECURSIVE APPROACH TO APPROXIMATE THE REGION UNDER ROC

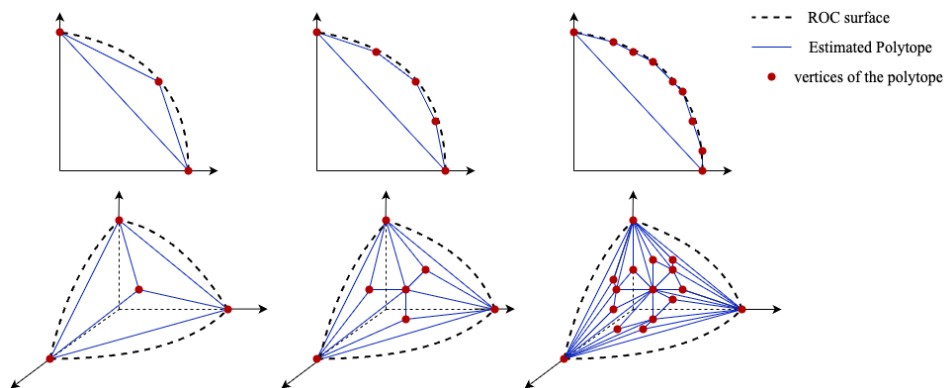

Figure 3: Estimation of 2D ROC curve (upper) and 3D ROC surface (bottom)

Consider the *derived outcome predictor* $\widetilde{Y}_{\boldsymbol{\theta}}$ defined in Definition 2.3, where each point on the ROC surface corresponds to the true positives achieved by $\widetilde{Y}_{\boldsymbol{\theta}}$ for different choices of $\boldsymbol{\theta}$. We use a recursive approach, illustrated in Fig. 3, to estimate the region under the ROC surface. We begin by selecting threshold vectors corresponding to predictors whose true positive point lies on one of the axes. These predictors are obtained by setting $\boldsymbol{\theta_1}$ to basis vectors (i.e., vectors with a single element equal to 1 and all others equal to zero). We then average these threshold vectors to obtain a new threshold, $\boldsymbol{\theta} = \frac{1}{N}\mathbf{1}_N$, which corresponds to the predictor that maximizes the model's accuracy. Note that we are only interested in the region above the hyperplane defined by $x + y + z + \cdots \geq 1$; any points below this region represent performance worse than that of a random classifier. The initial estimation of the region under the ROC surface is thus formed as a simplex, as shown in Fig. 2 (right). In the next step, for each pair of points (in 2D) or group of points forming an edge (in 2D) or a triangle (in higher dimensions) on the current simplex, we average their corresponding threshold vectors to generate new vertices. These new points are added recursively, progressively refining the approximation of the region under the ROC.

# I EXPERIMENTAL DETAILS

## I.1 DATA, MODELS, HYPERPARAMETER AND BASELINES

**Data, Models and Hyperparameter:** We provide the models and hyperparameters used for each dataset. All experiments were run on a local Linux server with a NVIDIA RTX 4070 GPU. The code is implemented in TensorFlow, simulating a FL setup with one server and multiple clients.

**Adult Dataset.** Each client's data is split into 60% training, 20% validation, and 20% testing. We use the `FedAvg` algorithm with $N = 2$ participating clients per round, local update epochs $E = 1$, and batch size $B = 512$. Local models are two-layer logistic regression networks (64 and 32 nodes) with ReLU activations, trained using Adam ($\eta = 0.001$).

**PublicCoverage Dataset.** Each client's data is split into 60% training, 20% validation, and 20% testing. The number of clients is $N = 50$, local update epochs is $E = 1$ and batch size is $B = 256$. The model architecture, training procedure, and evaluation follow those of the Adult dataset.

**HM10000 Dataset.** Data is split into 60% training, 20% validation, and 20% testing. Diagnostic classes are grouped into four categories: (1) pre-cancerous/cancerous (`akiec`, `bcc`, `mel`), (2) benign (`bkl`, `df`), (3) nevus-like (`nv`), and (4) vascular (`vasc`). Images are resized to $28 \times 28 \times 3$. Local models are CNNs with three convolutional layers (32, 64, 128 filters), followed by global average pooling and two dense layers (128 and 32 nodes). Models are trained with sparse categorical cross-entropy using Adam ($\eta = 0.0001$), with $E = 1$ and $B = 32$.

**Baselines:** We introduce the baselines used in the experimental section.

1. `Agnostic-FL` Mohri et al. (2019) improves client fairness through adversarial training, encouraging the model to perform well on the worst- performed client. The implementation follows https://github.com/YuichiNAGAO/agnostic_federated_learning

2. `q-FFL` (Li et al., 2019) enhances client fairness by minimizing an aggregated reweighted loss, parameterized by $q$, which prioritizes clients with higher local losses. We set $q = 4$ for the Adult dataset, following the original implementation, and $q = 1$ for HM1000.

3. `FCFL` (Cui et al., 2021) is designed to achieve local fairness and performance consistency through a constrained min-max optimization framework. We report results using their official implementation available at: https://github.com/cuis15/FCFL.

4. `FairFed` (Ezzeldin et al., 2023) is designed to achieve global fairness by adaptively adjusting the aggregation weights of different clients based on their local fairness metrics. The fairness budget parameter $\beta$ is set to 1 for both the Adult and ACSPublicCoverage datasets.

5. `Fair-Fate` (Salazar et al., 2023) aims to ensure global fairness by incorporating a momentum term to mitigate oscillations caused by fairness-agnostic gradients. Following (Salazar et al., 2023), we set the parameters $\{\lambda_0, \rho, \text{MAX}, \beta_0\}$ to $\{0.5, 0.05, 1, 0.99\}$ for the Adult dataset and $\{0.5, 0.05, 1, 0.9\}$ for ACSPublicCoverage.

6. `EquiFL` (Makhija et al., 2024) promotes both local and global fairness by adding a fairness-aware regularization term to the local loss. The regularization weight $w$ is set to $10^2$ for the Adult dataset and $10^5$ for ACSPublicCoverage.

7. `LOGO` Zhang et al. (2025)is also a post-processing approach but it only supports binary-class settings. It enforcing fairness by solving a bi-level optimization. The implementation follows https://github.com/liizhang/LoGofair

## I.2 PARETO FRONTIER OF THE ACCURACY-FAIRNESS TRADEOFFS

Fig. 4 presents the Pareto frontiers illustrating the trade-off between accuracy and each fairness concept. These results demonstrate that our framework can flexibly adjust fairness levels by modifying the fairness constraints in the linear program. Compared to the baselines, it achieves comparable accuracy across a range of fairness specifications.

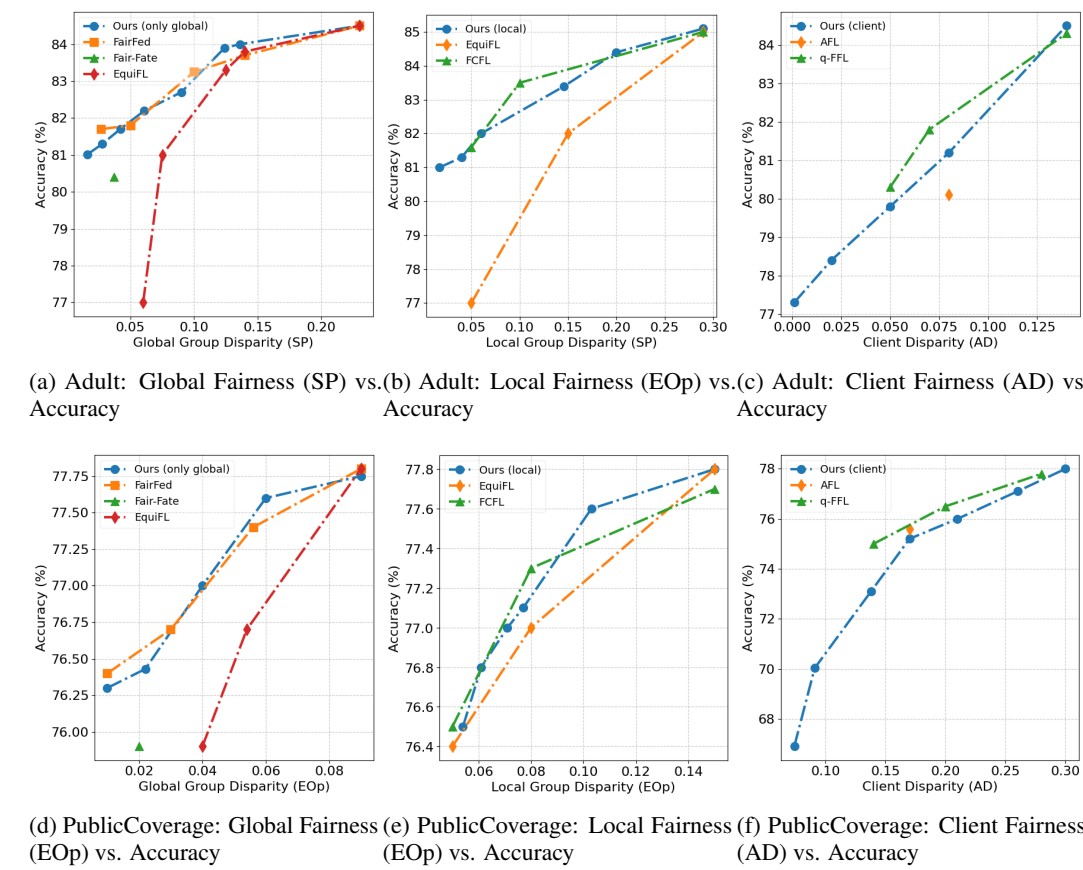

(a) Adult: Global Fairness (SP) vs. Accuracy

(b) Adult: Local Fairness (EOp) vs. Accuracy

(c) Adult: Client Fairness (AD) vs. Accuracy

(d) PublicCoverage: Global Fairness (EOp) vs. Accuracy

(e) PublicCoverage: Local Fairness (EOp) vs. Accuracy

(f) PublicCoverage: Client Fairness (AD) vs. Accuracy

Figure 4: Pareto Frontier of accuracy and each distributive fairness concept

## I.3 DATA HETEROGENEITY

We report the global, local, and client fairness achieved by our framework under varying levels of data heterogeneity. Results for scenario s1 and s5 are presented in Table 2. Results for intermediate scenarios s2–s4, where the sensitive attribute becomes increasingly imbalanced, are reported below.

Table 4: Local ($\Delta^l$), global group ($\Delta^g$) disparity, client disparity ($\Delta^c$) and accuracy (Acc) of our algorithm for multi-class tasks under s2-s4.

| Method | HM10000 (s2) | | | | HM10000 (s3) | | | | HM10000 (s4) | | | |
|---|---|---|---|---|---|---|---|---|---|---|---|---|
| | $\Delta^l$ | $\Delta^g$ | $\Delta^c$ | Acc | $\Delta^l$ | $\Delta^g$ | $\Delta^c$ | Acc | $\Delta^l$ | $\Delta^g$ | $\Delta^c$ | Acc |
| FedAvg | 0.26 | 0.29 | 0.05 | 81.1 | 0.37 | 0.21 | 0.06 | 80.6 | 0.42 | 0.21 | 0.08 | 81.2 |
| *Ours (all)* | 0.03 | 0.01 | 0.01 | 67.9 | 0.07 | 0.04 | 0.01 | 67.2 | 0.10 | 0.04 | 0.01 | 67.7 |
| *Ours (global)* | 0.36 | 0.05 | 0.12 | 76.3 | 0.43 | 0.01 | 0.18 | 75.4 | 0.41 | 0.02 | 0.30 | 75.0 |
| *Ours (local)* | 0.07 | 0.14 | 0.27 | 69.5 | 0.06 | 0.16 | 0.21 | 67.4 | 0.07 | 0.18 | 0.21 | 67.2 |
| *Ours (client)* | 0.43 | 0.10 | 0.02 | 76.9 | 0.41 | 0.10 | 0.01 | 76.9 | 0.31 | 0.07 | 0.01 | 74.0 |

## I.4 THE RELAXATION OF THE CONVEX PROGRAM

We conduct experiments on the UCI Adult dataset to illustrate the differences between the linear program (LP) that uses approximated ROC and the convex program (CP) with true ROC. All setups are the same as those in the paper. For the LP, we apply our framework with either local or global fairness constraints. For the CP based on ROC curves, global fairness is enforced by solving the CP

over the entire data distribution, while local fairness is enforced by solve the CP over each client's distribution. The results are shown in the following table.

Table 5: Accuracy, local, global group disparity and client disparity of post-processing using convex program (CP) and linear program (LP).

| Method | Adult (gender) | | | |
|---|---|---|---|---|
| | $\Delta_{\text{SP}}^{local}$ ($\downarrow$) | $\Delta_{\text{SP}}^{global}$ ($\downarrow$) | $\Delta_{\text{DM}}^{client}$ ($\downarrow$) | Acc ($\uparrow$) |
| FedAvg | 0.29 | 0.23 | $0.10 \pm 0.04$ | 84.9 |
| LP (only global) | 0.14 | 0.01 | 0.04 | 81.2 |
| CP (only global) | 0.11 | 0.01 | 0.06 | 82.3 |
| LP (only local) | 0.03 | 0.02 | 0.08 | 81.1 |
| CP (only local) | 0.02 | 0.01 | 0.06 | 81.8 |

The accuracy of the model under the LP (81.2%) and CP (82.3%) for enforcing global fairness differs by 1.1%. For enforcing local fairness, the accuracy under the LP (81.1%) and CP (81.8%) differs by 0.7%. These results indicate that the LP closely approximates the solution of the CP.

## I.5 DIFFERENTIAL PRIVACY

The statistics computed and transmitted by the client $c$, as described in Eq.(11) of the paper, are:

$$
\begin{aligned}
&\Pr_D(\widetilde{Y}_{\boldsymbol{\theta_1}} = y, Y = y, A = a \mid C = c) \\
&= \frac{\# \text{ of samples with } (\widetilde{Y}_{\boldsymbol{\theta_1}} = y, Y = y, A = a) \text{ in client } c}{\# \text{ of samples in client } c}, \\
&\Pr_D(Y = y, A = a \mid C = c) \\
&= \frac{\# \text{ of samples with } (Y = y, A = a) \text{ in client } c}{\# \text{ of samples in client } c}.
\end{aligned}
\tag{31}
$$

The statistics sent by the client $c$ after applying the Laplace Mechanism are:

$$
\begin{aligned}
&\Pr_D(\widetilde{Y}_{\boldsymbol{\theta_1}} = y, Y = y, A = a \mid C = c) + \text{Lap}(0|b_c), \\
&\Pr_D(Y = y, A = a \mid C = c) + \text{Lap}(0|b_c),
\end{aligned}
\tag{32}
$$

where $\text{Lap}(x|b) = \frac{1}{2b}e^{-\frac{|x|}{b}}$ is the density function of the Laplace distribution, and $b_c$ is the scale parameter of the Laplace distribution. The larger $b_c$, the greater the variance of the added noise.

The statistics will satisfy $\epsilon$-differential privacy (Dwork, 2006), if we set the scale parameter $b_c$ as:

$$
b_c = \frac{\Delta f_c}{\epsilon}
\tag{33}
$$

where $\Delta f_c$ represents the sensitivity for client $c$, which is the maximum difference in the statistics sent by the client when a single data point is added or removed.

In our setting, the sensitivity is:

$$
\Delta f_c = \frac{1}{\# \text{ of samples in client } c}.
\tag{34}
$$

Thus, for each client $c$, the scale parameter $b_c$ is given by:

$$
b_c = \frac{1}{(\# \text{ of samples in client } c) \cdot \epsilon}.
\tag{35}
$$

A larger $b_c$ corresponds to a smaller $\epsilon$, providing better privacy protection. To maintain the same level of privacy across all clients, clients with fewer samples will be given a larger $b_c$.

Local differential privacy (DP) in this section is applied during the communication of local statistics in step (2) of our training pipeline in Sec 4. to protect client-level privacy. Fig. 5 illustrates how DP mechanisms affect the fairness and accuracy of our algorithm on the *PublicCoverage* dataset. We apply the Laplace mechanism to the local statistics in Eq. equation 11, ensuring that they satisfy $\epsilon$-differential privacy.

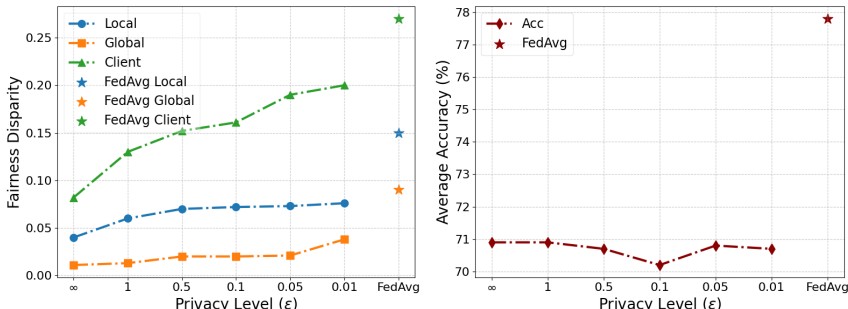

Figure 5: Global Group Disparity when enforcing local & client fairness under different data heterogeneity (right), $\epsilon$-Privacy vs. Different Fairness Concepts (left)

Compared to the `FedAvg` with $\epsilon = 0.01$, our framework reduces local disparity by 50% $(0.150 \to 0.076)$, global disparity by 58% $(0.09 \to 0.038)$, and client disparity by 30% $(0.27 \to 0.20)$. These results demonstrate the effectiveness of our framework in mitigating all fairness concepts under a 0.01-differentially private setting. As $\epsilon$ decreases (i.e., privacy protection becomes stronger), local, global and client disparities tend to increase, which shows the trade-off between privacy and fairness under our framework.

## J  THE USE OF LARGE LANGUAGE MODELS (LLMS)

This paper uses LLMs to check grammar and spelling in the writing.

