# OpenReview forum: "Post-Processing Approach for Distributive Fairness in Multi-Class Federated Learning"
_ICLR.cc/2026/Conference — Submitted to ICLR 2026_

### Official Review · Reviewer_tAYg · 2025-10-25

**Soundness:** 2
**Presentation:** 3
**Contribution:** 2
**Rating:** 4
**Confidence:** 3

**Summary:**

This paper introduces a framework for multi-class FL that balances various combinations of distributive fairness, which is a pioneer work to that seeks to optimize global group, local group, and individual fairness collectively.

**Strengths:**

1. The theoretical analysis of this paper is solid.

2. This work is the first to propose a unified framework that seeks to optimize global group, local group, and individual fairness collectively.

**Weaknesses:**

1. There are only 2 clients for the experiments on Adult, and 5 clients for the experiments on HM10000. Note that in real-world FL settings, the number of clients could range from 100 to 10000. Although the paper uses 50 clients for the ACS dataset, this is only one special case not a common setting throughout the experimental evaluation.

2. The paper lacks evaluation on real-world large-scale vision/text datasets.

3. The assumption that each client has access to all sensitive groups is not very practical. Also, there may be incompatibility between client fairness, local fairness, and global group fairness.

**Questions:**

Please refer to the weakness part.

---

> ### Author Response · Authors · 2025-11-25
>
> **Concern 1 (w1,w2): Results on large-scale text datasets with 100-1000 clients.**
>
> Our framework is a post-processing approach that can be applied to data of any modality and to pretrained models. The results show that our framework enforces fairness with lower communication and computation costs. We believe that the main conclusions remain consistent regardless of which datasets are used in the experiments.
>
> We add preliminary experiments on the large-scale text dataset BiasBios. The task is to predict the occupation (|$\mathcal{Y}$| = 28\) of individuals from their raw-text biographies. The data are mined from the Common Crawl corpus, and gender is the sensitive attribute. We use the version of BiasBios in [1], which contains 393,423 examples in total. We manually split the data across 300 clients: for the first 100 clients, the majority of samples are female; for the next 100 clients, 50% of the samples are female; and for the final 100 clients, the majority of samples are male. For each clients, we split the data as training set (0.4), validation set (0.3) and test set (0.3).  We fine-tune BERT
> models from the bert-base-uncased checkpoint [7]. Then apply our post-processing technique. Results on Table 1 shows our framework effectively reduces the global, local and client disparity.
>
>
>
> | Framework | local | global | client | Acc |
> | :---: | :---: | :---: | :---: | :---: |
> | FedAvg | 0.183 | 0.141 | 0.172 | 86.2 |
> | Ours(all) | 0.003 | 0.001 | 0.016 | 83.1 |
> | Ours(local) | 0.002 | 0.004 | 0.120 | 84.4 |
> | Ours (global) | 0.201 | 0.002 | 0.086 | 85.3 |
> | Ours (client) | 0.191 | 0.013 | 0.003 | 84.0 |
> Table 1: The performance of post-processing w.r.t multi-class Equal Opportunity on BiasBios
>
> **Concern 2 (W3 part1): The assumption that each client has access to all sensitive groups is not very practical.**
>
> Knowing group identity is the basis for measuring and enforcing group fairness. This paper examines group fairness, particularly as required by legal and regulatory standards. The 80\% rule in the Disparate Impact Law, for example, states: "A selection rate for any race, sex, or ethnic group that is less than four‑fifths (80\%) of the rate for the group with the highest rate will generally be regarded by federal enforcement agencies as evidence of adverse impact." This reflects statistical parity, where the positive rate between sensitive and non-sensitive groups should differ by no more than 0.2. Without knowing an individual's sensitive attributes, it is difficult for an organization to measure and enforce fairness. This is the reason why, when we discuss fairness, most literature [2-5] discuss and address fairness under knowledge of group identity. There are some work such as [6] that do not assume access to sensitive attributes, but those work focus on making accuracy consistent across subpopulations. They do not enforce and explore fairness according to legal or regulatory metrics such as Statistical Parity and Equal Opportunity.
>
> **Concern 3 (W3 part2 ): Regarding the incompatibility between client fairness, local fairness, and global group fairness.**
>
> Global fairness, local fairness and client fairness can coexist in FL. We formally prove that both the convex program and its LP relaxation are always feasible, which proves that there always exists a classifier that satisfies all three fairness notions simultaneously.
>
> Besides that, we can provide some simple classifier examples to show that there are always exist models that can achieve three fairness concepts.
>
> (1) Consider a classifier, $
> \tilde{Y}: \mathcal{X} \times \mathcal{A} \times \mathcal{C} \rightarrow \mathcal{Y}
> $
> that satisfies
> \begin{equation}
>     {\rm Pr}_D\big(\widetilde{Y} = y \mid Y = y, A = a, C = c\big) = c,
>     \quad \forall\, y \in \mathcal{Y},\, a \in \mathcal{A},\, c \in \mathcal{C},
> \end{equation}
> for some $c \in [0,1]$. Those predictor whose true positives are the same across all clients, group and class can easily build by our algorithm. Such a predictor $\widetilde{Y}$ satisfies local, global, and client fairness with respect to multi-class Equal Opportunity.
>
> (2) Consider a naive predictor that outputs $\widetilde{Y}$ with constant probability $c$, independent of $X, A, C, Y$. This classifier satisfies all three fairness notions under Statistical Parity.
>
> We report experimental results with all fairness tolerances set to \(0\) on Table 2 of the paper and Table 1 in this rebuttal.  In those settings, disparities in global group fairness, local group fairness, and client fairness are all reduced to (near) zero. Together, the theoretical analysis and empirical evidence demonstrate that these fairness metrics can be satisfied simultaneously.
> Moreover, given a fairness level $\epsilon$, our algorithm can return a model that satisfies the fairness constraints while maximizing model accuracy.

---

> ### Author Response · Authors · 2025-11-25
> **References**
>
> [1].Shauli Ravfogel, Yanai Elazar, Hila Gonen, Michael Twiton, and Yoav Goldberg. Null It Out:
> Guarding Protected Attributes by Iterative Nullspace Projection. In Proceedings of the 58th
> Annual Meeting of the Association for Computational Linguistics, 2020.
>
> [2]. Moritz Hardt, Eric Price, and Nati Srebro. Equality of opportunity in supervised learning. Advances
> in neural information processing systems, 29, 2016.
>
> [3].Ninareh Mehrabi, Fred Morstatter, Nripsuta Saxena, Kristina Lerman, and Aram Galstyan. A
> survey on bias and fairness in machine learning. ACM computing surveys (CSUR), 54(6):1–35,
> 2021
>
> [4].Han Zhao and Geoffrey J Gordon. Inherent tradeoffs in learning fair representations. Journal of
> Machine Learning Research, 23(57):1–26, 2022.
>
> [5].Yahya H Ezzeldin, Shen Yan, Chaoyang He, Emilio Ferrara, and A Salman Avestimehr. Fairfed:
> Enabling group fairness in federated learning. In Proceedings of the AAAI conference on artificial
> intelligence, volume 37, pages 7494–7502, 2023
>
> [6]. Afroditi Papadaki, Natalia Martinez, Martin Bertran, Guillermo Sapiro, and Miguel Rodrigues.
> Federated fairness without access to sensitive groups. arXiv preprint arXiv:2402.14929, 2024.
>
> [7]. Jacob Devlin, Ming-Wei Chang, Kenton Lee, and Kristina Toutanova. BERT: Pre-training of Deep
> Bidirectional Transformers for Language Understanding. In Proceedings of the 2019 Conference of
> the North American Chapter of the Association for Computational Linguistics: Human Language
> Technologies, volume 1, 2019

---

### Official Review · Reviewer_Un6i · 2025-10-30

**Soundness:** 3
**Presentation:** 2
**Contribution:** 3
**Rating:** 6
**Confidence:** 4

**Summary:**

This paper studies the problem of fairness in federated learning, considering three definitions: global group fairness, local group fairness, and client fairness. The question is what is the maximum achievable accuracy under various combinations of fairness, i.e., all three, any two, or just one. The paper also proposes a post-processing algorithm to obtain a model with near-optimal accuracy while satisfying pre-specified fairness constraints. Experimental results are included on Adult, ACSPublicCoverage and HM10000 datasets.

**Strengths:**

-- New formulation considering three types of fairness simultaneously. While local and global fairness have been looked at, including their tradeoffs via a convex optimization, the introduction of client fairness into the formulation is quite interesting.

-- Leads to a nice convex optimization to find the optimal accuracy under constraints on the different fairness criteria.

-- Proposes a post-processing algorithm that further tries to achieve the optimal accuracy. The algorithm describes the role of each client in the execution of the strategy, showing how it will be implemented in a distributed setting.

-- Experiments are provided on 3 datasets, and multiple baselines have been considered. Adult has 2 clients while PublicCoverage has 50 clients.

**Weaknesses:**

-- Figures could be better for comparing the tradeoffs rather than tables. There are some figures in the Appendix, but the captions/legends were not very clear to understand how it outperforms SOTA.

-- It is a bit confusing to say it outperforms SOTA. It seems they achieve better fairness by paying a cost in accuracy (unless I am misunderstanding the table)? The benefit will be clearer in a tradeoff plot. (communication/computation benefit is acknowledged)

-- Another option is to consider a radar chart to better understand the performance comparison.

-- Some grammar issues and typos were noted.

125. (1) We formally defines
129. We defines

A few others as well.

**Questions:**

Q1. Could you explain how and in what aspect these algorithms outperform SOTA (other than communication/computation benefit)? Or, is there a better way to visualize the accuracy-fairness tradeoff if there is one?

Q2. The paper has discussed extensions to some other group fairness metrics in the Appendix. I was wondering if it is possible to define a general class of fairness metrics over which this kind of a convex optimization formulation can be extended?

Q3. This is optimal for post-processing techniques, but is it possible that there may exist other in-processing/pre-processing techniques that could lead to an even better tradeoff?

Q4. Are there any assumptions that the initial FedAvg solution needs to satisfy?

---

> ### Author Response · Authors · 2025-11-25
> **Concern 1(W1, W2, Q1): Regarding how our framework outperforms SOTA**
>
> **Concern 1(W1, W2, Q1): Regarding how our framework outperforms SOTA:**
>
> We compare our framework with all baselines from the following perspectives: (1) communication cost, (2) computational cost, (3) the flexibility of adjusting fairness levels, (4) the ability to enforce multiple fairness concepts, (5) the accuracy cost of enforcing fairness. In addition to lower computational and communication costs, our framework allows precise and flexible control of fairness levels by tuning $\epsilon$ in the linear program (LP). In contrast, other baselines require extensive trial-and-error over different hyperparameters to reach a desired fairness level. Those framework are less flexible for adjusting fairness levels. Most baselines only enforce single fairness concept. Our framework can enforce various combinations of distributive fairness, i.e., all three, any two, or just one, depending on the application. Therefore, when comparing with these methods, we run our framework with only one fairness constraint. We tune all baselines so that they achieve a similar level of fairness and then compare accuracy. Under this setting, our framework is not always the best in terms of performance, but it achieves competitive accuracy. Overall, our framework outperforms SOTA in terms of (1) communication cost, (2) computational cost, (3) flexibility in adjusting fairness levels, and (4) the ability to enforce multiple fairness concepts. It achieves competitive accuracy under fairness constraints compared with baseline methods.
>
> **Concern 2 (Q2): General class of fairness metrics:**
>
> Our post-processing framework can enforce fairness w.r.t metrics that can be formulated as linear constraints over each client’s confusion matrix or score distribution. The fairness metrics discussed in this paper, including Statistical Parity, Equal Opportunity, Equalized Odds and Mistreatment Disparity, can all be expressed as linear constraints applied to each client’s confusion matrix.
>
> **Concern 3 (Q3): Regarding the better fairness-accuracy tradeoffs:**
>
> Yes, it is possible. First, this paper formally answers the question: 'What is the best accuracy we can expect for a given level of local, global, and client fairness?' We propose a convex program that answers the question. The objective value of the convex program is the maximum achievable accuracy under the fairness constraints. However, in practice, the computational complexity of estimating the feasible region of the convex program increases exponentially with respect to the number of classes. This paper uses an LP to approximate the solution of the convex program. So, the accuracy returned by our training pipeline is near-optimal. It is possible that some algorithms achieve better accuracy. However, in-processing and pre-processing methods have two problems: first, they cannot balance the three fairness concepts, and second, they come with large computational and communication costs. If we disregard computational complexity, solving the convex program yields the best accuracy compared to other algorithms.
>
> **Concern 4 (Q4): Regarding the assumptions we made**
>
> The analysis of the maximum achievable accuracy under fairness constraints assumes that the score function is the Bayes-optimal score, as defined in Definition 2.2. The score represents the probability that a given sample belongs to each class. The score function can be empirically approximated by minimizing the multi-class cross-entropy loss.

---

### Official Review · Reviewer_Cory · 2025-10-31

**Soundness:** 3
**Presentation:** 3
**Contribution:** 3
**Rating:** 4
**Confidence:** 4

**Summary:**

The paper proposes a post-processing framework for federated learning that enforces three distributive fairness notions—global group fairness, local group fairness, and client fairness—while selecting an outcome predictor that maximizes accuracy under explicit constraints.   It extends ROC-based post-processing from binary to multi-class settings by defining a multi-class ROC surface, characterizing the region under this surface via supporting hyperplanes, and proving that this region is convex and contains all achievable true-positive vectors, which yields a convex program for optimal accuracy under fairness constraints. Experimental result on three datasets reduces global/local/client disparity with competitive accuracy.

**Strengths:**

- The paper addresses a timely challenge in federated learning, reducing global, local, and client disparity with competitive accuracy.

- Clear formalization of three fairness notions in multi-class FL with precise constraints for global, local, and client fairness.

- Empirical efficiency: consistent reductions in three disparities with fewer communication rounds compared to baselines.

- The paper is well organized and easy to read.

**Weaknesses:**

- The theoretical framework presented in this paper is nearly identical to that proposed in [1], substantially diminishing its technical contribution and novelty. I would be willing to raise my score if the authors can clearly differentiate their approach and address this concern.

- Local statistics may reveal important properties of the local datasets in high-stakes scenario, even without containing any per-user information. Differential privacy alone is insufficient to prevent such leakage. The authors ought to consider encrypted computation on the server side.

- Accuracy cost can be substantial, especially when enforcing all three notions on some datasets. The Pareto frontier is referenced but not visualized in the main text.

[1] The Cost of Local and Global Fairness in Federated Learning, AISTATS 2025.

**Questions:**

1. Why does the theoretical approach in this paper appear overly similar to that of [1]? Is this essentially the same method applied to a different problem setting?

2. How sensitive are results to Bayesian score miscalibration?

---

> ### Author Response · Authors · 2025-11-25
> **Concern 1 (W1,Q1): Regarding the difference between this paper and [1]**
>
> This paper addresses open challenges in FL fairness that are not covered in [1] or the current literature. Without the theoretical framework in [1], these problems could not be solved. The paper also investigates technical issues arising from framework in [1], which we discuss in what follows.
>
> ➤The first open problem is to explore the **interplay between local, global and client fairness**. As discussed in lines 67 to 79 and Appendix A, real-world applications often involve multiple stakeholders, each of whom require fairness from different perspectives. These fairness concepts may be aligned, where enforcing one naturally improves another, or they may not, where enforcing one does not guarantee the others, depending on how data is distributed across clients.
> While several works enforce a single fairness concept, there is limited understanding of how local, global, and client fairness relate to one another under different levels of data heterogeneity. Understanding this interplay is essential for identifying when stakeholders’ interests align and when they diverge.
> This paper leverages the technique developed in [1] to explore this interplay. Our results show that (as discussed in line  429-449) enforcing local fairness improves global fairness when data are i.i.d across clients. However, this improvement diminishes as the sensitive attribute becomes increasingly imbalanced across clients. Enforcing client fairness enhances global group fairness when the sensitive attribute is completely skewed across clients, but this improvement declines as the sensitive attribute becomes less correlated with client identity.
>
> ➤ The second challenge is **enforcing common global fairness metrics, such as statistical parity (SP), equal opportunity (EOp) and equalized odds (EO), when the sensitive attribute is completely skewed across clients.** This situation is common in real-world applications. Racial segregation is one example, where different racial groups live in different districts. Similarly, different local hospitals may serve demographically distinct populations.
> When the sensitive attribute is distributed across different clients, the client identity aligns with the sensitive attribute. Client fairness align the group fairness. Existing work that enforce client fairness ensures consistent accuracy across clients. These methods cannot enforce standard fairness metrics such as SP, EOp, or EO. Prior approaches [2,3], for example, use dynamic weighting to assign greater weights to clients with lower performance, or employ adversarial training to solve a min-max optimization problem. These frameworks reduce accuracy disparities, but they cannot enforce fairness in terms of SP, EOp, or EO, nor can they be easily adapted to do so.
> On the other hand, methods that enforces group fairness cannot enforce these metrics when the sensitive attribute is completely skewed across clients. FairFed [4], for example, computes dynamic aggregation weights based on the mismatch between global fairness measurements, calculated from the overall dataset, and local fairness measurements at each client. The algorithm prioritizes clients whose local fairness aligns more closely with the global fairness target. However, if some clients contain only privileged groups and others only unprivileged groups, the local fairness disparity has no value and their framework becomes ineffective.
> Overall, existing methods cannot enforce fairness when the sensitive attribute is skewed across clients, which limits their applicability in real-world scenarios.
>
> Achieving fairness when the sensitive attribute is completely skewed across clients and understanding the interplay between fairness concepts are fundamental questions for fairness in FL. These problems are not answered in the initial paper by [1]. Besides that, we make the following technical contributions upon [1].
>
> ➤ The framework uses a linear program to approximate the solution of a convex program. We study how the relaxation of the convex program into a linear program impacts fairness and model accuracy. The results show that, while under the same fairness level, the accuracy obtained by the convex program is slightly better, but the linear program closely approximates the optimal accuracy, with the accuracy difference is $\leq 0.02$.
>
> ➤This paper discusses a different approach for approximating the region under the ROC surface. While [1] uses a simplex to approximate it, this paper proposes a recursive approach to approximate it. Details of the recursive approach are provided in Appendix H.
>
> ➤ The cost-of-fairness analysis assumes the score function is optimal (Def. 2.2), but in practice we only approximate the optimal score. We bound the resulting accuracy gap between models from Bayes-optimal and near-Bayes-optimal scores in terms of the miscalibration error of the Bayes-optimal score (as suggested by the reviewer; see the reply to Concern 4).

---

> ### Author Response · Authors · 2025-11-25
> **Concern 2 (W2): Regarding the privacy of Local statistics**
>
> We thank the reviewer for raising concerns regarding client-level statistical privacy. While we have considered encrypted computation techniques such as Secure Aggregation, it is not applicable to our framework. This is because the linear program solved by the server depends on specific statistics from each individual client (e.g., true positives), rather than aggregated statistics across clients. Secure Aggregation is primarily designed for scenarios that require only aggregated information, which does not align with our needs.
>
> Another possible direction is the use of Homomorphic Encryption, which enables computations to be performed directly on encrypted data. This would allow the server to solve the linear program using encrypted parameters and then return an encrypted solution to be decrypted by the client. Some prior work such as [5] has considered solving linear programs using cryptographic protocols. Such techniques could potentially be applied in our framework to further protect client-level privacy.
>
> **References**
>
> [1].  Duan, Yuying, Gelei Xu, Yiyu Shi, and Michael Lemmon. "The cost of local and global fairness in Federated Learning." In International Conference on Artificial Intelligence and Statistics, pp. 4186-4194. PMLR, 2025.
>
> [2]. Mehryar Mohri, Gary Sivek, and Ananda Theertha Suresh. Agnostic federated learning. In Inter-
> national conference on machine learning, pages 4615–4625. PMLR, 2019.
>
> [3]. Tian Li, Maziar Sanjabi, Ahmad Beirami, and Virginia Smith. Fair resource allocation in federated
> learning. arXiv preprint arXiv:1905.10497, 2019.
>
> [4]. Yahya H Ezzeldin, Shen Yan, Chaoyang He, Emilio Ferrara, and A Salman Avestimehr. Fairfed:
> Enabling group fairness in federated learning. In Proceedings of the AAAI conference on artificial
> intelligence, volume 37, pages 7494–7502, 2023
>
> [5].Octavian Catrina and Sebastiaan De Hoogh. Secure multiparty linear programming using fixed-
> point arithmetic. In European Symposium on Research in Computer Security, pages 134–150.
> Springer, 2010.

---

> > ### Author Response · Authors · 2025-11-25
> > **Concern 3 (W3): Accuracy cost can be substantial**
> >
> > We agree that accuracy cost can be substantial for some FL setup. Our framework formally answers the question: What is the best accuracy we can expect for a given level of local, global and client fairness?
> > The inherent cost of fairness is algorithm-independent, that is, no matter what algorithm you use, the accuracy cannot be better than the solution given by convex program (6).
> > It is true that the accuracy cost can be substantial for some FL setup. With our framework,  users can clearly see what they will sacrifice for fairness. This allows users to decide whether the cost is acceptable, or if they would prefer to relax the fairness constraints as much as allowed. Our framework gives the truth and knowing the truth never hurts.
> >
> > 'The Pareto frontier is referenced but not visualized in the main text.'  We will move the Pareto frontier illustrating the fairness–accuracy tradeoffs from the appendix to the main text.

---

> ### Author Response · Authors · 2025-11-25
> **Concern 4 (Q4):  How sensitive are results to Bayesian score miscalibration?**
>
> We provide a simple analysis of the accuracy of the fair outcome predictor when the score is miscalibrated.
>
> First, this paper proves that when the score $R: \mathcal{X} \times \mathcal{A} \times \mathcal{C} \rightarrow[0,1]^N$ is Bayes optimal, that is, its components $R(x, a, c)=\left[r_1(x, a, c), r_2(x, a, c), \ldots, r_N(x, a, c)\right]$ satisfy:
>
> $$
> \forall y \in\{1,2, \ldots, N\}, \quad r_y(x, a, c)=\operatorname{Pr}_D(Y=y \mid X=x, A=a, C=c),
> $$
>
> the accuracy given by our convex program is the maximum achievable accuracy under the imposed
> fairness constraints.
>
> Next, we consider the case where the given score is the near-optimal Bayesian Score and analyze the accuracy of the corresponding fair outcome predictor.
>
> For each client $c$, group $a$, and class $y$, we define the miscalibration error $\epsilon_{a c}^y$ as the expected value of the difference between the Bayes optimal score function $r_y$ and the near-optimal Bayes score function $\hat{r}_y$ :
>
> $$
> \epsilon_{a c}^y=\mathbb{E}_D\left[\left|r_y(x, a, c)-\hat{r}_y(x, a, c)\right| \cdot \mathbf{1}(A=a, C=c, Y=y)\right] .
> $$
>
> Let $\epsilon$ denote the largest miscalibration error across all clients, groups and classes:
> $
> \epsilon = \max_{a,y,c} \epsilon_{ac}^y
> $
>
> The fair outcome predictor is derived from the score function and the coefficients $\beta_{ac}^y$ (for $a \in \mathcal{A}$, $c \in \mathcal{C}$, $y \in \mathcal{Y}$), as shown in Algorithm 1, where these coefficients are determined by the solution of the linear program. Let $\widetilde{Y}$ be the fair outcome predictor obtained from the Bayes-optimal scores $r_y(x,a,c)$ for $y \in \mathcal{Y}$ with coefficients $\beta_{ac}^y$, and let $\widehat{Y}$ be the fair outcome predictor derived from the scores $\widehat{r_y}(x,a,c)$ for $y \in \mathcal{Y}$ with coefficients $\hat{\beta}_{ac}^y$.
> Let:
>
> $$
> \alpha = \max_{y,a,c} \big|\beta_{ac}^y - \hat{\beta}_{ac}^y\big|.
> $$
>
> where $\alpha$ captures the sensitivity of the LP with respect to its constraints. When the score function is miscalibrated, the feasible region of the LP changes, which in turn changes its solution. $\alpha$ is LP-parameter dependent and can be explored by sensitivity analysis for LP [6].
>
> Let $p_{ac}^y = \Pr_D(A=a, C=c, Y=y)$ and let $\operatorname{TP}_{ac}^y(\cdot)$ denote the true positive rate for class $y$, group $a$, and client $c$ under a given predictor. Using the $0\text{-}1$ loss $\ell$, the difference in accuracy between $\widetilde{Y}$ and $\widehat{Y}$ can be bounded as:
>
> $$
> \left|\mathbb{E}_D[\ell(\tilde{Y}(X, A, C), Y)]-\mathbb{E}_D[\ell(\widehat{Y}(X, A, C), Y)]\right|
> $$
>
> $$=|\sum_{a, y, c} p_{a c}^y \mathrm{TP}_{a c}^y(\tilde{Y}) - \sum\_{a,y,c} p\_{a c}^y \mathrm{TP}\_{a c}^y(\widehat{Y})| $$
>
> $$
> \leq \sum\_{a, y, c} p\_{a c}^y\left|\mathrm{TP}\_{a c}^y(\tilde{Y})-\mathrm{TP}\_{a c}^y(\widehat{Y})\right|
> $$
>
> $$
> \leq \max\_{a, y, c}\left|\mathrm{TP}\_{a c}^y(\widetilde{Y})-\mathrm{TP}\_{a c}^y(\widehat{Y})\right|
> $$
>
> $$
> =\max\_{a, y, c}\left|\beta\_{a c}^y+\beta\_{a c}^0 \mathrm{TP}\_{a c}^y\left(\widetilde{Y}\_\theta\right)-\hat{\beta}\_{a c}^y-\hat{\beta}\_{a c}^0 \mathrm{TP}\_{a c}^y\left(\widehat{Y}\_\theta\right)\right| \leq \alpha(1+\epsilon) .
> $$
>
>
>
>
>
> In particular, when the score is exactly Bayesian optimal, both $\alpha$ and $\epsilon$ are equal to zero. The above bound shows that the difference in accuracy is determined by the miscalibration error $\epsilon$ and the sensitivity $\alpha$ of the LP solution.
>
>
> **References**
>
> [6]. Linear Programming, Sensitivity Analysis, and Related Topics. (Prentice Hall, 2010)

---

### Meta-Review · Area_Chair_WZGh · 2026-01-06

**Summary:**

Some of the major comments include
1. Novelty compared to [1] is limited
2. privacy concerns
3. Insufficient evaluations

**Reviewer Concerns:**

Novelty compared to [1] not sufficiently addressed. Some more experimental results have been given.

**Reviewer Scores:**

Unchanged

---

### Decision · Program_Chairs · 2026-01-26

Reject